

SciPost Phys. Lect. Notes 43 (2022)

# Sub-GeV dark matter models and direct detection

**Tongyan Lin**⋆

University of California San Diego, La Jolla, CA, USA

⋆ tongyan@physics.ucsd.edu

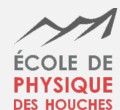

*Part of the Dark Matter*
*Session 118 of the Les Houches School, July 2021*
*published in the Les Houches Lecture Notes Series*

## Abstract

Lecture notes for Les Houches Summer School 2021: Dark Matter. These lectures give a brief introduction to sub-GeV dark matter models, and then reviews theory and approaches to direct detection of sub-GeV dark matter.

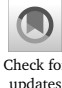

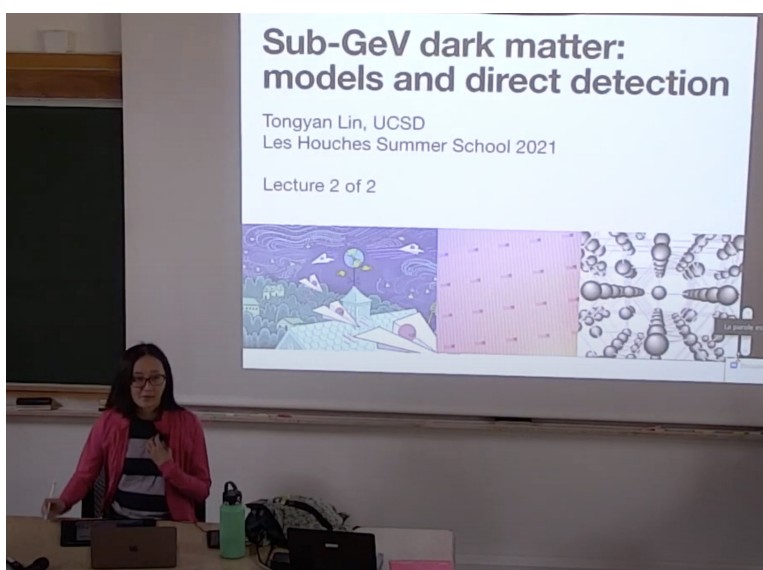

# Contents

These lecture notes are largely based on material from my TASI lecture notes from 2019 [1] (Secs. 1,4) and from a review article written with Yonatan Kahn [2] (Secs. 2,3,4). The material has been lightly edited and adapted for the format and length of the two lectures planned at the Les Houches summer school; interested readers are encouraged to follow up with those reviews for original material and more in-depth discussion. In these notes, as in the reference material, we use natural units with $\hbar = c = 1$.

# 1 Overview of sub-GeV models

Our starting point is thermal relic dark matter, which has historically driven extensive searches for WIMPs. Recall that the full DM relic abundance is obtained when there is an annihilation cross section given by:

$$\langle \sigma v \rangle \simeq \frac{\sqrt{g_*}}{g_{*,S}} \frac{10}{\text{eV} \times M_{\text{pl}}} \simeq \frac{1}{10^9 \text{ GeV}^2}, \tag{1}$$

which is the *minimum* annihilation cross section needed for a thermal relic DM candidate, in order to avoid an overabundance. We will take as an illustrative example annihilation that occurs through an *s*-channel mediator with mass $m_V$ (remaining for the moment agnostic as to the identity of $V$):

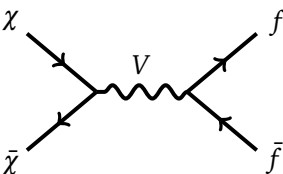

where the vector $V$ has coupling $g_\chi$ with the DM and coupling $g_f$ with the final state fermions.

We will neglect the mass of the final state fermions. In the non-relativistic limit, the cross section for this process is given by

$$\sigma = \int d\Omega_{\text{cm}} \frac{|\mathbf{p}_f|}{16\pi^2 E_{\text{cm}}^3 |\mathbf{v}_1 - \mathbf{v}_2|} |\mathcal{M}|^2 = \int d\Omega_{\text{cm}} \frac{1}{|\mathbf{v}_1 - \mathbf{v}_2|} \frac{|\mathcal{M}|^2}{32\pi^2 s} , \qquad (2)$$

where $\Omega_{\text{cm}}$ are center of mass scattering angles, the center of mass energy is $s = E_{\text{cm}}^2 = 4m_\chi^2 + O(m_\chi T) + ...$, and we used that $|\mathbf{p}_f| \approx E_{\text{cm}}/2$ in the limit of massless fermions $f$. Using this result, we can approximate the thermally averaged $\langle \sigma v \rangle$ for annihilation by

$$\langle \sigma v \rangle \simeq \frac{|\mathcal{M}|^2}{32\pi m_\chi^2}. \qquad (3)$$

Assuming Dirac fermion DM, a single flavor/color of the fermion, and a vector mediator, the spin-averaged matrix element squared of the process is given by

$$|\mathcal{M}|^2 \approx g_\chi^2 g_f^2 \frac{32 m_\chi^4}{(s - m_V^2)^2} \qquad (4)$$

in the nonrelativistic limit.

- $m_V > m_\chi$: In this case, the heavy $V$ state generates a four-fermion interaction with amplitude $g_\chi g_f / m_V^2$. The annihilation cross section can be estimated as

$$\langle \sigma v \rangle \simeq \frac{16\pi \alpha_\chi \alpha_f m_\chi^2}{m_V^4} , \qquad (5)$$

  with $\alpha_\chi \equiv g_\chi^2/(4\pi)$ and $\alpha_f \equiv g_f^2/(4\pi)$.

- $m_V < m_\chi$: We find that the annihilation $\chi\bar{\chi} \to f\bar{f}$ is:

$$\langle \sigma v \rangle \simeq \frac{\pi \alpha_\chi \alpha_f}{m_\chi^2}. \qquad (6)$$

However, a new process is then kinematically allowed, $\chi\bar{\chi} \to VV$. If $m_V \ll m_\chi$ then the only mass scale in the problem is $m_\chi$ and we obtain:

$$\langle \sigma v \rangle_{\chi\bar{\chi} \to VV} \simeq \frac{\pi \alpha_\chi^2}{m_\chi^2}. \qquad (7)$$

If $\alpha_\chi \gg \alpha_f$, then the relic abundance may be primarily determined by this latter process. The "secluded" scenario is where $\alpha_\chi \gg \alpha_f$ and where $V$ is a new mediator not already present in the Standard Model [3]. Then we may regard the $\chi$ and $V$ states as comprising a "dark sector", and couplings to the SM thermal bath are not important for thermal freezeout.

We find, generally, that the thermally-averaged cross section is bounded from above by

$$\langle \sigma v \rangle \lesssim \frac{\pi \max(\alpha_\chi \alpha_f, \alpha_\chi^2)}{m_\chi^2} , \tag{8}$$

since for the case $m_V > m_\chi$ there is an additional suppression in the cross section by $(m_\chi/m_V)^4$. (An exception is if there is a resonance in the $s$-channel, $m_V \approx 2m_\chi$.) If the desired cross section is that of Eq. 1, then there are a few important lessons to draw from this. First, one can set an upper bound on the DM mass, if we assume perturbative couplings – this is also known as a perturbative unitarity bound [4]. Taking $\alpha_{\chi,f} \to 1$ in Eq. 8 then gives $m_\chi \lesssim 50 - 100$ TeV. The second lesson is that a TeV-scale DM candidate is a viable possibility, where $\alpha_{\chi,f}$ can be on the order $10^{-2} - 10^{-1}$. Indeed, the WIMP miracle is the observation that $\langle \sigma v \rangle$ above can be rewritten in the form $\langle \sigma v \rangle \approx \alpha_w^2/\text{TeV}^2$ with $\alpha_w \approx 0.03$ for $SU(2)_L$ weak interactions. A minimal WIMP model comes from introducing an extended electroweak multiplet; if the lightest state in the multiple has zero electric charge, this provides a good DM candidate. These candidates have mass at the TeV scale and above; for a recent systematic study of such candidates, see [5].

In turning to lighter DM below the weak scale, the minimalist approach becomes insufficient at a certain point. Let us take $V$ to be a weak scale mediator, which could be the Higgs or an EW gauge boson. We have a cross section given by Eq. 5, which can be rewritten as

$$\langle \sigma v \rangle \approx \frac{m_\chi^2}{\text{GeV}^2} \frac{\alpha_\chi \alpha_f}{\alpha_w^2} \frac{1}{10^9 \, \text{GeV}^2} , \tag{9}$$

where we have taken $m_V \approx 100$ GeV. If the couplings are the weak gauge couplings, $\alpha_\chi \alpha_f = \alpha_w^2$, then the cross section drops below the desired thermal relic cross section when $m_\chi < $ GeV. Weak interactions would thus lead to an overabundance of sub-GeV DM, a conclusion commonly known as the Lee-Weinberg bound [6]. The implication is that for sub-GeV DM, *new mediators below the weak scale are required.*

However, there is one more candidate for $V$ in the Standard Model, the photon. Since $m_V \ll m_\chi$, the Lee-Weinberg bound does not apply. Although this would appear to go against the notion of dark matter as "dark", one must work through this possibility in a more quantitative way. Consider a DM candidate which has a small, fractional electric charge $Q \ll 1$, otherwise known as a millicharged or minicharged DM candidate.[1] The annihilation cross section $\bar{\chi}\chi \to f\bar{f}$ for a single charged species $f$ is given by $\langle \sigma v \rangle \approx \pi \alpha_{\text{em}}^2 Q^2/m_\chi^2$; this should be summed over all final states with appropriate final state charges and color factors. Neglecting such factors for the purpose of estimation, one finds that

$$Q \simeq 10^{-3} \left( \frac{m_\chi}{\text{GeV}} \right) , \tag{10}$$

in order to obtain the desired thermal relic cross section. Such charges can potentially be constrained in accelerator experiments, as well as in stellar environments such as SN1987a. One of the strongest test of charged DM arises from its behavior in the early universe. At redshifts $z \gtrsim 1000$, the universe was mostly ionized in the form of free protons and electrons. A millicharged DM candidate can scatter off the protons and electrons with a Rutherford-type cross section, leading to both a suppression of the growth of DM structure [10] as well as a DM-baryon drag force which leaves an imprint on the CMB anisotropies [11–15]. This leads to a strong bound on $Q$, which excludes the charges in Eq. 10. Thus, a model for sub-GeV DM which obtains its relic abundance by thermal freezeout generically requires additional

---

[1]For a discussion of specific models realizing this, see for example Refs. [7–9].

new sub-GeV states for sufficient DM annihilation. Note however that charged DM is still a possible DM candidate if it is produced by freeze-in, which we will return to in Sec. 1.1.1.

The arguments here motivate the study of dark sectors for sub-GeV DM, that contain both the DM and other light states. The excess of energy and entropy density in a dark sector may be excluded by other cosmological considerations, requiring that the excess be deposited back into the SM thermal bath. As a result, we will often consider new light *mediators* to the SM.

## 1.1  Dark photons

Perhaps the most often studied mediator in recent times is the kinetically-mixed dark photon, owing to the appeal of a simple $U(1)$ extension with a rich phenomenology and the absence of any flavor problems. Focusing on the interactions with just the photon (most relevant for the low-energy phenomenology), the vacuum interactions for this portal are:

$$\mathcal{L} \supset -\frac{1}{4}F_{\mu\nu}F^{\mu\nu} - \frac{1}{4}V_{\mu\nu}V^{\mu\nu} + \frac{\varepsilon}{2}F_{\mu\nu}V^{\mu\nu} + \frac{1}{2}m_V^2 V_\mu V^\mu + eA_\mu J_{\rm EM}^\mu + g_\chi V_\mu J_D^\mu \,, \qquad (11)$$

where $J_{\rm EM}^\mu$ is the electromagnetic current and $J_D^\mu$ is a dark current with gauge coupling $g_\chi$. We denote the electron charge throughout as $e = \sqrt{4\pi\alpha}$ with $\alpha \simeq 1/137$ the fine structure constant[2]. The kinetic mixing parameter $\varepsilon$ can be positive or negative, though constraints are typically shown on the absolute value $|\varepsilon|$. In addition, there are the kinetic or mass terms for any dark charged particles, which aren't written explicitly in order to be general. The vector mass could arise from a dark Higgs mechanism, where the dark Higgs boson is extremely massive and has been integrated out of the theory (i.e, a Stueckelberg mass term). However, in some dark sector models the dark Higgs is also a light degree of freedom and important to the phenomenology.

First consider the case that $m_V = 0$. Then we can make a field redefinition $\tilde{V}_\mu = V_\mu - \varepsilon A_\mu$. This eliminates the kinetic mixing term, and we are left with the following interactions:

$$\mathcal{L} \supset -\frac{1}{4}(1-\varepsilon^2)F_{\mu\nu}F^{\mu\nu} - \frac{1}{4}\tilde{V}_{\mu\nu}\tilde{V}^{\mu\nu} + eA_\mu J_{\rm EM}^\mu + g_\chi \left(\tilde{V}_\mu + \varepsilon A_\mu\right)J_D^\mu \,, \qquad (12)$$

where the factor of $(1-\varepsilon^2)$ in the photon kinetic term can be eliminated by a field (or electric charge) redefinition. In the absence of a dark current $J_D^\mu$, we would have a completely decoupled vector $\tilde{V}$ with no observable effects. Hence in the massless vector limit, we expect that the only limits would come from effects that involve the DM. Now suppose there is a DM particle, for example $J_D^\mu = \bar{\chi}\gamma^\mu\chi$. In this basis, it is clear that the DM couples to the photon with an effective charge $\varepsilon g_\chi$ or millicharge $Q = \varepsilon g_\chi / e$. This model gives an explicit realization of millicharged DM, discussed earlier in these lectures around Eq. 10.

Now we examine what happens when $m_V \neq 0$, starting with the vacuum Lagrangian. Another often-used basis comes from making the field redefinition $\tilde{A}_\mu = A_\mu - \varepsilon V_\mu$, which eliminates the kinetic mixing term:

$$\mathcal{L} \supset -\frac{1}{4}\tilde{F}_{\mu\nu}\tilde{F}^{\mu\nu} - \frac{1}{4}(1-\varepsilon^2)V_{\mu\nu}V^{\mu\nu} + \frac{1}{2}m_V^2 V_\mu V^\mu + e(\tilde{A}_\mu + \varepsilon V_\mu)J_{\rm EM}^\mu + g_\chi V_\mu J_D^\mu \,, \qquad (13)$$

where we see that dark photon mass eigenstate $V$ couples to SM charged particles. Of course, the physics is independent of any field redefinitions or change of basis for the $A, V$ fields[3]. This

---

[2]Note that we are using Heaviside-Lorentz conventions for the electric charge as is common in high-energy physics, where $\alpha = e^2/(4\pi)$. This differs by factors of $4\pi$ from cgs-Gaussian units where $\alpha = e^2$.

[3]One could also use this basis for the massless $m_V$ limit. Then in the absence of a dark current, there is no phenomenological difference from regular QED. One can then check that Coulomb scattering, bremsstrahlung etc, are all the same as in QED up to an overall redefinition of electric charge squared as $e^2(1+\varepsilon^2)$ in the limit $\varepsilon \ll 1$. This redefinition of electric charge is identical to the charge renormalization in the basis of Eq. 12.

basis is most often used in collider studies of dark photon phenomenology, where the vacuum assumption is valid. However, one must be more careful when considering dark photons in a dense medium, for example in the early universe, in stars, or in a solid state material.

When $m_V > m_\chi$, then direct annihilation of DM DM $\to e^+ e^-$ is sufficient to set the relic abundances. In this direct coupling scenario, DM with mass $m_\chi \gtrsim$ few MeV is viable given constraints from Big Bang Nucleosynthesis (BBN). This is because primordial element abundances would be sensitive to additional particles in equilibrium with the SM thermal bath at the time of nucleosynthesis [16]. If $m_V < m_\chi$, DM annihilation to mediators is possible; then the DM mass may be as low as the 1-10 keV scale, which is consistent with warm DM and BBN limits as long as the dark sector is decoupled from the SM plasma and sufficiently cold. And in the limit of ultralight mediator $m_V \ll$ eV, the DM behaves effectively like a millicharged particle and freeze-in of DM through an ultralight vector is an interesting benchmark, as we discuss next.

### 1.1.1 Freeze-in

There is another well-studied benchmark particularly relevant for direct detection of sub-GeV dark matter, which strictly speaking is not a thermal candidate. Freeze-in [17] is a mechanism whereby rare interactions within the SM thermal bath slowly build up an abundance of DM. (In the usual freeze-in story, it is thus assumed that dark sector particles are not produced at an appreciable level through decay of the inflaton during reheating.) As a specific example, let's look at freeze-in by $s$-channel annihilation of SM particles, such as $e^+ e^-$, into DM particles. The coupling of the DM particles is assumed to be sufficiently feeble, that the reaction is never in equilibrium.

In the "UV-dominated" scenario, the production cross section depends on a high scale $\Lambda$; for example, for scalar DM the interaction is a dimension-5 operator which we can parameterize as $\frac{g_\chi g_e v_H}{\Lambda^2} \chi^2 \bar{e} e$. We have included the factor of the Higgs vacuum expectation value, $v_H = 246$ GeV, to account for the fact that the operator is not $SU(2)_L$ invariant. Then the cross section goes as

$$\langle \sigma v \rangle \simeq \frac{\alpha_\chi \alpha_e v_H^2}{\Lambda^4}. \tag{14}$$

The rate of producing a DM particle per electron is $\Gamma_{e^+ e^- \to \chi\chi} = n_e \langle \sigma v \rangle \sim \alpha_\chi \alpha_e T^3 v_H^2 / \Lambda^4$ for $T \gg m_e, m_\chi$. Thus, the number of DM particles created per electron in a Hubble time is $\Gamma H^{-1}$. The abundance of total newly-created DM at any given time is

$$Y_\chi = \frac{n_\chi}{s} \simeq \frac{n_e \Gamma H^{-1}}{s} \simeq \frac{\Gamma}{g_{*,S} H} \simeq \frac{\alpha_\chi \alpha_e v_H^2 M_{\text{pl}} T}{\sqrt{g_*} g_{*,S} \Lambda^4}, \tag{15}$$

with the greatest abundance produced at the highest $T$ (as long as $T < \Lambda$). The relic density is sensitive to the reheating of the universe and the maximum available temperatures. Assuming that this process gives all of the dark matter and requiring that the highest $T >$ MeV and $m_\chi >$ keV gives a lower bound on $\Lambda/(\alpha_\chi \alpha_e)^{1/4} \gtrsim 10^6$ GeV. With such high scales or small couplings, the prospects for laboratory detection in the near future are quite limited.

On the other hand, if the mediator is lighter than the DM, then we can have "IR-dominated" freeze-in. Going back to the example introduced above Eq. 2 where DM is coupled to a light vector mediator, the production cross section for $e^+ e^- \to \chi \bar{\chi}$ has the form

$$\langle \sigma v \rangle \simeq \frac{\alpha_\chi \alpha_e}{T^2} \tag{16}$$

in the limit of $m_V \ll m_\chi$. Similar to above, the comoving abundance of total newly-created DM is given by

$$Y_\chi = \frac{n_\chi}{s} \approx \frac{\Gamma}{g_{*,S} H} \simeq \frac{\alpha_\chi \alpha_e M_{\rm pl}}{\sqrt{g_*} g_{*,S} T} \,, \tag{17}$$

so that most of the DM is produced at around the lowest $T$ where the process is kinematically accessible. Either the DM becomes too heavy ($T < m_\chi$) which suppresses the production rate, or the electrons become too dilute ($T < m_e$). While the total abundance should be obtained by integrating the production at all times, we can estimate the relic abudance by taking the comoving abundance at freeze out to be $Y_{\rm fo} = Y(T)$ at the lowest $T$. There are two possibilities depending on the DM mass, which results in the following condition on the couplings

$$\alpha_\chi \alpha_e \simeq \begin{cases} \sqrt{g_*} g_{*,S}|_{T=m_\chi} \times \frac{\rm eV}{M_{\rm pl}} \approx 3 \times 10^{-27} - 10^{-26}, & m_\chi > m_e \\ \sqrt{g_*} g_{*,S}|_{T=m_e} \times \frac{\rm eV}{M_{\rm pl}} \frac{m_e}{m_\chi} \approx 3 \times 10^{-27} \times \frac{m_e}{m_\chi}, & m_\chi < m_e \end{cases}. \tag{18}$$

If we take the mediator to be the SM photon, this is then an example of a sub-GeV DM candidate that does not require any new mediators beyond the SM! In particular, the couplings required satisfy the bound on $Q$ discussed below Eq. 10. (Note that for $m_\chi < m_e$ and the SM photon as the mediator, there is an additional large production mechanism whereby the in-medium plasma oscillations can decay to $\chi\bar{\chi}$; this modifies the coupling constants above by about an order of magnitude, depending on the mass [9].) Despite the tiny couplings, this scenario is potentially detectable with direct detection or indirect searches when the mediator mass is much smaller than the DM mass.

### 1.1.2 Example benchmarks

Here we identify some specific benchmark parameters that will motivate our discussion of direct detection searches. To be concrete, consider the following two benchmarks with the kinetically-mixed vector portal:

- $m_\chi = 10$ MeV, $m_V = 30$ MeV, $g_\chi = 0.3$ and $\varepsilon = 10^{-4}$ (thermal relic, direct coupling),

- $m_\chi = 1$ MeV, $m_V = 10^{-12}$ eV, $g_\chi = 3 \times 10^{-6}$, and $\varepsilon = 10^{-6}$ (freeze-in),

where $g_\chi$ is the coupling of the dark photon with Dirac fermion DM.

★ *Exercise:* Check that the first benchmark is a good candidate for a thermal relic, that the entire relic abundance can be produced by freeze-in for the second benchmark, and that the above parameters can satisfy the existing constraints on dark photons.

The spin-averaged matrix element squared and cross section for DM-electron scattering can be written as

$$|\mathcal{M}|^2 = \frac{16 g_\chi^2 \varepsilon^2 e^2 m_\chi^2 m_e^2}{((q_\mu^2) - m_V^2)^2} \approx \frac{16 g_\chi^2 \varepsilon^2 e^2 m_\chi^2 m_e^2}{(|\mathbf{q}|^2 + m_V^2)^2}. \tag{19}$$

$$\bar{\sigma}_e \equiv \frac{16\pi \mu_{\chi e}^2 \varepsilon^2 \alpha_\chi \alpha}{((\alpha m_e)^2 + m_V^2)^2} \,, \tag{20}$$

where $\alpha$ is the fine structure constant, $m_e$ is the electron mass, $\mu_{\chi e}$ is the DM-electron reduced mass, and $\alpha_\chi = g_\chi^2/(4\pi)$. $|\mathbf{q}|$ is the momentum transfer, and a typical value for electron scattering is $|\mathbf{q}| \simeq \alpha m_e$. Interestingly, the two benchmarks give comparable values for $\bar{\sigma}_e \approx 10^{-37} {\rm cm}^2$. Consider for example a target with atomic Xe, in which DM can excite the

outer 6 valence shell electrons in each atom. We can estimate the rate as $R_\chi \sim N_T \frac{\rho_\chi}{m_\chi} \bar{\sigma}_e v$ with the number of targets per kilogram as $N_T \sim 6 \times 6 \times 10^{26}/(131)$ and typical velocity $v \sim 10^{-3}$, giving $R_\chi \sim 10^{-3}$ events/kg/s or 300 events/kg/day. Therefore, cosmologically motivated benchmarks could in principle be detectable even with relatively small detectors of around 1 kg that are sensitive to single electron ionizations. The same is true for detectors sensitive to single phonon excitations as well, though in that case getting a rate estimate is more involved, as we must know something about phonons.

## 1.2 Scalar mediators

The dark photon model illustrates the "top-down" approach, where we began with a particular model of DM dynamics in the early universe to derive DM interactions in the laboratory. That approach predicts a particular coupling strength of DM to electron and proton number density in the nonrelativistic limit. Another approach one might take is to start with a general scalar or vector mediator coupling to electron, proton, or neutron number density. Consider DM that couples to nucleons or electrons only, mediated by a scalar Yukawa interaction. Let's parameterize the interactions as $y_n \phi \bar{n} n$ (where we assume equal coupling to neutrons and protons) and $y_e \phi \bar{e} e$, where $y_n$ and $y_e$ need not be directly related. The DM also has an interaction $y_\chi \phi \bar{\chi} \chi$.

For this benchmark, the cosmology is quite different from the dark photon model. Due to the strong stellar and fifth force constraints on a scalar mediator, thermal freezeout of DM in the direct coupling scenario is highly constrained, and there are strong upper bounds on the possible scattering rates. For sub-GeV DM coupling to electrons, thermal freezeout from $\chi \bar{\chi} \to e^+ e^-$ (though an off-shell $\phi$) is only viable for $m_\chi, m_\phi \gtrsim 10\,\text{MeV}$ due to BBN constraints. Even then, there are strong constraints from SN1987a and beam dump constraints on such a mediator coupled to electrons. Thermal freezeout in a secluded sector is possible for lower mass DM, but the combination of stellar constraints and self-interacting DM constraints leads to strong upper bounds on $y_n y_\chi$ and therefore on $\bar{\sigma}_e$, which are not detectable with any proposed DM-electron scattering experiments [18, 19].

For sub-GeV DM coupling to nucleons, the annihilation process $\chi \bar{\chi} \to n \bar{n}$ is clearly not possible when the temperature of the universe is well below $T \approx \text{GeV}$, while at higher temperatures, one needs to specify a microscopic coupling of DM to quarks or gluons, which can be model-dependent. In addition, there are strong constraints on mediators coupling to quarks or gluons from observations of rare meson decays. The upshot is that thermal freezeout scenarios with sharp benchmark values of $y_n y_\chi$ are excluded for sub-GeV dark matter [20]. Again, one can consider enlarging the dark sector, which does lead to viable thermal relic possibilities. Combining astrophysical/cosmological and laboratory constraints still leads to upper bounds on $y_n y_\chi$ and therefore on $\bar{\sigma}_n$ [21]. For MeV-GeV mass dark matter and the massless mediator limit ($m_\phi \ll m_\chi v_0$), the bounds on $\bar{\sigma}_n$ are the weakest, with the potential for large signals in direct detection experiments. However, there are more stringent limits on sub-MeV DM and the massive mediator limit [18–20]. Despite these caveats, we will use this as an example model in considering DM-phonon interactions, since it is the simplest interaction to consider in that context.

## 2 Direct detection of sub-GeV dark matter

At this point in the school, we have discussed direct detection of WIMPs with nuclear recoils, including how to determine DM scattering rates $dR/dE_R$ in terms of the DM interaction and velocity distribution. In order to treat scattering for WIMPs into other types of modes, including

condensed matter systems, it is useful to introduce an alternative writing of the DM scattering rate which will generalize more easily.

## 2.1 Scattering rate

In an arbitrary detector of volume $V$ and density $\rho_T$, Fermi's Golden Rule gives the scattering rate for DM per unit target mass:

$$R_\chi = \frac{1}{\rho_T} \frac{\rho_\chi}{m_\chi} \int d^3\mathbf{v} f_\chi(\mathbf{v}) \frac{V d^3\mathbf{p}'_\chi}{(2\pi)^3} \sum_f |\langle f, \mathbf{p}'_\chi | \Delta H_{\chi T} | i, \mathbf{p}_\chi \rangle|^2 2\pi \delta(E_f - E_i + E'_\chi - E_\chi), \quad (21)$$

where $f_\chi(\mathbf{v})$ is the lab-frame DM velocity distribution, $\Delta H_{\chi T}$ is the non-relativistic Hamiltonian governing the interactions between DM and the target constituents, and $|i\rangle$, $|f\rangle$ are the initial and final detector states with energies $E_i$ and $E_f$ respectively. We assume that the DM interactions with the target $\Delta H_{\chi T}$ may be treated as a perturbation on the free-particle DM Hamiltonian, such that unperturbed eigenstates are plane waves $|\mathbf{p}\rangle$, and that there is no entanglement between the DM and the target so that $|i, \mathbf{p}_\chi\rangle \equiv |i\rangle \otimes |\mathbf{p}_\chi\rangle$ and similarly for $|f, \mathbf{p}'_\chi\rangle$. To simplify the expression further, we assume that the matrix element also factorizes into Fourier components $\mathbf{q}$ as

$$\langle f, \mathbf{p}'_\chi | \Delta H_{\chi T} | i, \mathbf{p}_\chi \rangle \equiv \int \frac{d^3\mathbf{q}}{(2\pi)^3} \langle \mathbf{p}'_\chi | \mathcal{O}_\chi(\mathbf{q}) | \mathbf{p}_\chi \rangle \times \langle f | \mathcal{O}_T(\mathbf{q}) | i \rangle \quad (22)$$

$$= \frac{1}{V} \sqrt{\frac{\pi \bar{\sigma}(q)}{\mu_\chi^2}} \langle f | \mathcal{O}_T(\mathbf{q}) | i \rangle, \quad (23)$$

where the $\mathcal{O}_\chi$ and $\mathcal{O}_T$ operators only add on the DM and target system states, respectively. In the second line, we have inserted plane wave states for the DM, e.g., $e^{i\mathbf{p}_\chi \cdot \mathbf{r}}/\sqrt{V}$, and used the fact that the matrix element $\langle \mathbf{p}'_\chi | \mathcal{O}_\chi(\mathbf{q}) | \mathbf{p}_\chi \rangle$ will lead to momentum conservation with $\mathbf{q} \equiv \mathbf{p}_\chi - \mathbf{p}'_\chi$. The quantity $(\pi \bar{\sigma}(q)/\mu_\chi^2)^{1/2}$ (where $q \equiv |\mathbf{q}|$) corresponds to the strength of the interaction potential in terms of a cross section $\bar{\sigma}(q)$ and mass parameter $\mu_\chi$, and we will give examples later for particular models. With this convention, $\mathcal{O}_T$ is a dimensionless operator and while it only acts on the target system, it could still depend on the DM model, such as the strength of DM coupling to the electron, proton, neutron constituents of the system.

To continue our factorization of the DM and target system portions of the above rate, we can introduce an auxiliary variable $\omega$ and integrate over $\omega$ with a delta function $\delta(\omega + E'_\chi - E_\chi)$. This gives the rate as

$$R_\chi = \frac{1}{\rho_T} \frac{\rho_\chi}{m_\chi} \int d^3\mathbf{v} f_\chi(\mathbf{v}) \int \frac{d^3\mathbf{q}}{(2\pi)^3} d\omega \frac{\pi \bar{\sigma}(q)}{\mu_\chi^2} \delta(\omega + E'_\chi - E_\chi)$$

$$\times \underbrace{\frac{2\pi}{V} \sum_f |\langle f | \mathcal{O}_T(\mathbf{q}) | i \rangle|^2 \delta(E_f - E_i - \omega)}_{S(\mathbf{q}, \omega)}. \quad (24)$$

Note that we can swap between $\mathbf{q}$ and $\mathbf{p}'_\chi$ using momentum conservation for the DM, but that we have not assumed momentum conservation in the target; as we will see, a crystal explicitly breaks translation invariance in a number of ways, so the eigenstates of the target are not necessarily momentum eigenstates. Eq. 24 gives a factorized form of the rate, where all of the dynamics of the target system are in contained in the final terms of the expression. This target response piece is called the *dynamic structure factor*, and denoted $S(\mathbf{q}, \omega)$. The factor of $1/V$ is included in the normalization to indicate that we are dealing with an intrinsic quantity

(since the sum over final states also scales as $V$) rather than an extrinsic quantity. As noted above, the target response does still depend on details of the DM model. To obtain the rate, the target response is weighted by the DM potential strength, and integrated over the phase space in terms of momentum transferred $\mathbf{q}$ and energy deposited $\omega$ by the DM, as well as the DM velocity distribution. We will use this form of the rate throughout.

At this point, the only assumption we have made about the target system is that it can be treated with non-relativistic quantum mechanics. The approach can thus be applied to nuclear recoils or condensed matter systems. We also generally work with systems in the ground state at zero (or at least negligible) temperature, so that we do not have to sum over an ensemble of initial states, and we will use $|i\rangle$ and $|0\rangle$ interchangeably to refer to the initial (ground) state.

## 2.2 Interaction strength

We can connect the quantity $\bar{\sigma}(q)$ with particle physics models discussed in Sec. 1. In the non-relativistic limit, the dark photon model yields the following interaction Hamiltonian between DM and charged particles, to leading order in the relative velocity:

$$\Delta H_{\chi Q} = \int \frac{d^3\mathbf{q}}{(2\pi)^3} e^{i\mathbf{q}\cdot(\mathbf{r}_Q - \mathbf{r}_\chi)} \frac{\varepsilon Q e g_\chi}{q^2 + m_V^2}, \tag{25}$$

where $\mathbf{r}_\chi$ is the DM position operator, $\mathbf{r}_Q$ is the position operator of a particle of electric charge $Qe$, $e = \sqrt{4\pi\alpha}$ is again the electron charge, and $g_\chi$ is the dark charge. Evaluating between plane-wave DM states gives

$$\langle \mathbf{p}'_\chi | \Delta H_{\chi Q} | \mathbf{p}_\chi \rangle = \int \frac{d^3\mathbf{q}}{(2\pi)^3} \frac{d^3\mathbf{r}_\chi}{V} e^{i(\mathbf{p}_\chi - \mathbf{p}'_\chi)\cdot\mathbf{r}_\chi} e^{i\mathbf{q}\cdot(\mathbf{r}_Q - \mathbf{r}_\chi)} \frac{\varepsilon Q e g_\chi}{q^2 + m_V^2} = \frac{1}{V} \frac{\varepsilon Q e g_\chi}{q^2 + m_V^2} e^{i\mathbf{q}\cdot\mathbf{r}_Q}, \tag{26}$$

where in the last equality the integration over the DM coordinate enforces momentum conservation, $\mathbf{q} = \mathbf{p}_\chi - \mathbf{p}'_\chi$. The matrix element of $\Delta H_{\chi Q}$ thus has the factorized form of Eq. 23, with

$$\langle f, \mathbf{p}'_\chi | \Delta H_{\chi Q} | i, \mathbf{p}_\chi \rangle = \frac{1}{V} \frac{\varepsilon Q e g_\chi}{q^2 + m_V^2} \langle f | e^{i\mathbf{q}\cdot\mathbf{x}_Q} | i \rangle \equiv \frac{1}{V} \mathcal{V}(q) \langle f | e^{i\mathbf{q}\cdot\mathbf{x}_Q} | i \rangle, \tag{27}$$

where we identify the cross section $\bar{\sigma}(q)$ as propoortional to the scattering potential $\mathcal{V}(q)$,

$$\bar{\sigma}(q) = \frac{\mu_{T\chi}^2}{\pi} \left( \frac{\varepsilon Q e g_\chi}{q^2 + m_V^2} \right)^2 \equiv \frac{\mu_{T\chi}^2}{\pi} (\mathcal{V}(q))^2 \tag{28}$$

and $\mu_{T\chi} = \frac{m_T m_\chi}{m_T + m_\chi}$ is the DM-target reduced mass; for a target proton or electron, for instance, $m_T = m_p$ or $m_e$ respectively.

It is common in the DM literature to rewrite $\bar{\sigma}(q) = \bar{\sigma}_T F_{\mathrm{DM}}^2(q)$, where $\bar{\sigma}_T$ is a fiducial cross section at fixed momentum transfer $q_0$,

$$\bar{\sigma}_T = \frac{\mu_{T\chi}^2}{\pi} \left( \frac{\varepsilon Q e g_\chi}{q_0^2 + m_V^2} \right)^2 \tag{29}$$

and $F_{\mathrm{DM}}(q)$ is a momentum-dependent *DM form factor*

$$F_{\mathrm{DM}}(q) \equiv \frac{q_0^2 + m_V^2}{q^2 + m_V^2}, \tag{30}$$

which parameterizes the momentum dependence of the scattering potential. For $T = e$, $\bar{\sigma}_T$ can be interpreted as a cross section for DM scattering off a free electron at a reference momentum

$q_0$, which is typically taken to be the inverse Bohr radius, $q_0 = 1/a_0 = \alpha m_e \simeq 3.7$ keV. For $T = p$, $\bar{\sigma}_p$ is the DM-proton cross section and $q_0$ is an arbitrary reference momentum which is often taken to be $q_0 = m_\chi v_0$. The two limits $F_{\mathrm{DM}} \to 1$ and $F_{\mathrm{DM}} \to (q_0/q)^2$ correspond to a *heavy mediator*, $m_V \to \infty$, or *light mediator*, $m_V \to 0$, respectively. Since the mass of the dark photon is unknown, these two limiting cases span the range of possibilities for the scattering amplitude. In position space, the heavy mediator limit corresponds to a contact interaction, $\mathcal{V}(\mathbf{r}_\chi - \mathbf{r}_Q) \propto \delta^{(3)}(\mathbf{r}_\chi - \mathbf{r}_Q)$.

Plugging in some numerical values, we find that for the freezeout scenario with $m_V > m_\chi$, the typical electron cross section is

$$\bar{\sigma}_e \simeq 3 \times 10^{-39} \mathrm{cm}^2 \left( \frac{10 \text{ MeV}}{m_\chi} \right)^2, \tag{31}$$

independent of $m_V$, $g_D$, and $\varepsilon$. Assuming a typical electron density of $n_e = 10^{24}/\mathrm{cm}^3$ in a generic material, the mean free path of a 10 MeV DM particle in a generic detector is

$$\lambda = (n_e \bar{\sigma}_e)^{-1} \simeq 4 \times 10^{12} \text{ m}. \tag{32}$$

Unlike ordinary Coulomb scattering between charged SM particles, then, there is no possibility of multiple scattering in any detector (or even of correlating scattering events between two nearby detectors on an event-by-event basis); thermal relic DM experiments are rare-event searches.

Motivated by the search for WIMP-nucleus scattering, the other case we will consider in this review is DM that couples to protons and neutrons only, mediated by a Yukawa interaction. The DM-nucleon Hamiltonian is given by

$$\Delta H_{\chi n} = \int \frac{d^3 \mathbf{q}}{(2\pi)^3} \, e^{i\mathbf{q} \cdot (\mathbf{r}_n - \mathbf{r}_\chi)} \frac{y_n y_\chi}{q^2 + M^2} \tag{33}$$

for a mediator of mass $M$, where $n$ denotes either a proton or a neutron. The coupling $y_n$ now plays the role of the charge of a nucleon with respect to this new mediator, and $y_\chi$ is the DM coupling. We will assume equal proton and neutron coupling for simplicity. As before, we can define a DM-nucleon fiducial cross section

$$\bar{\sigma}_n = \frac{\mu_{\chi n}^2}{\pi} \left( \frac{y_n y_\chi}{q_0^2 + M^2} \right)^2, \tag{34}$$

where $q_0 = m_\chi v_0$ as before. Again, there is also a DM form factor $F_{\mathrm{DM}}(q)$, which is identical to Eq. 30 but with the replacement $m_V \to M$.

## 2.3 Kinematics

Suppose incoming DM with momentum $\mathbf{p}_\chi = m_\chi \mathbf{v}$ scatters off a detector target and exits with momentum $\mathbf{p}'_\chi$. Using that for nonrelativistic DM, the energy eigenstates of the DM Hamiltonian are $E_\chi = p_\chi^2/2m_\chi$ and $E'_\chi = p'^2_\chi/2m_\chi$, we may write the energy deposited in the target in terms of the momentum transfer $\mathbf{q}$:

$$\omega_\mathbf{q} = E_\chi - E'_\chi = \frac{1}{2} m_\chi v^2 - \frac{(m_\chi \mathbf{v} - \mathbf{q})^2}{2m_\chi} = \mathbf{q} \cdot \mathbf{v} - \frac{q^2}{2m_\chi}. \tag{35}$$

Eq. (35) defines the kinematically-allowed region in $\omega, \mathbf{q}$ for DM scattering as a function of DM mass and velocity.[4] As shown in Fig. 1, for fixed $\mathbf{v}$, this region is bounded by an inverted

---

[4]For bosonic DM, there is the additional possibility of *absorption*, where the entire mass-energy of the DM is transferred to the target, yielding $\mathbf{q} = m_\chi \mathbf{v}$ and $\omega \approx m_\chi$. Condensed matter systems then provide sensitivity to eV-mass DM and below. While we focus exclusively on the scattering process in this review, see Refs. [22–31] for a dedicated treatment of absorption in various targets.

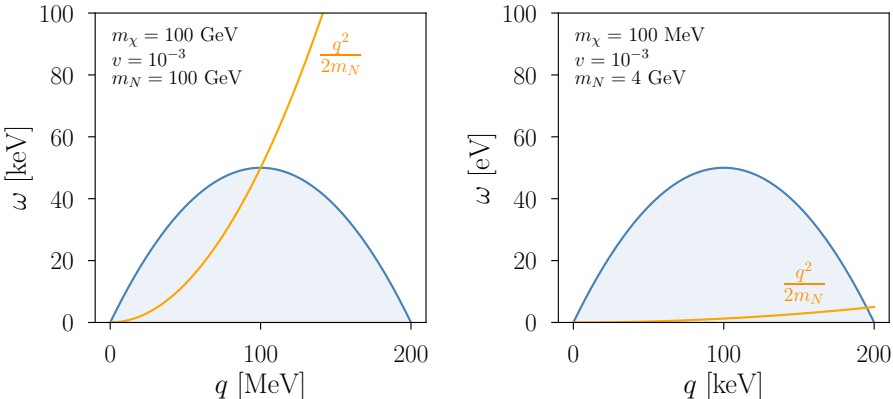

Figure 1: The blue shaded region is the kinematically allowed region for DM scattering with $v = 10^{-3}$. This region must be matched onto the allowed response of the target system, which for nuclear recoils is a resonant response at $q^2/(2m_N)$. When $m_\chi = m_N$ we have the optimal matching since it is possible for DM to deposit up to all of its initial kinetic energy. When $m_\chi \ll m_N$, there is nonzero response of the system only in a corner of the phase space, and the maximum recoil energy is much smaller.

parabola in the $\omega - q$ plane; as $v$ increases, the parabola moves up in $\omega$ since the DM has more kinetic energy. The upper boundary of the parabola corresponds to forward scattering, $\mathbf{q} \cdot \mathbf{v} = qv$, which gives the largest possible energy deposit $\omega$ for a given $q$. The apex of the parabola corresponds to $q = m_\chi v$ and $\omega = \frac{1}{2} m_\chi v^2$, where the target absorbs all of the kinetic energy of the incoming DM and $\mathbf{p}'_\chi = 0$. The right boundary of the parabola corresponds to maximum momentum transfer for a given energy deposit, which reduces to elastic "brick-wall" scattering when $\mathbf{p}'_\chi = -\mathbf{p}_\chi$ and $\omega_{\mathbf{q}} \to 0$.

This allowed scattering region must be compared with the allowed response of a target system in terms of $\mathbf{q}, \omega$. For instance, for nuclear recoils the dynamic structure factor is

$$S(\mathbf{q}, \omega) = 2\pi n_{\mathrm{nuc}} |F_N(q)|^2 \delta\left(\omega - \frac{q^2}{2m_N}\right) \tag{36}$$

for a nucleus with mass number $A$ and number density $n_{\mathrm{nuc}}$. $|F_N(q)|^2$ is the elastic recoil form factor, where in the low-momentum limit $|F_N(q)|^2 \to A^2$. This means we have a resonant response of the system at the nuclear recoil dispersion relation. This resonant response must them be compared with the allowed scattering phase space. Defining $\cos\theta = \hat{\mathbf{v}} \cdot \hat{\mathbf{q}}$, setting $\omega = \frac{q^2}{2m_N} = \omega_{\mathbf{q}}$ gives:

$$qv \cos\theta = \frac{q^2}{2\mu_{\chi N}} \,, \tag{37}$$

where $\mu_{\chi N} = m_\chi m_N / (m_\chi + m_N)$ is the reduced mass for the DM-nucleus system. There is a maximum momentum transfer $q_{\mathrm{max}} = 2\mu_{\chi N} v$. For a WIMP scattering off a typical nucleus, then $\mu_{\chi N} \simeq 10 - 100\,\mathrm{GeV}$ and $q_{\mathrm{max}} \simeq 20 - 200\,\mathrm{MeV}$. The corresponding maximum recoil energy is

$$E_R^{\mathrm{max}} = \frac{2\mu_{\chi N}^2 v^2}{m_N} \simeq 20 - 200\,\mathrm{keV}. \tag{38}$$

However, we see that for $m_\chi \ll m_N$, $E_R^{\max} \approx 2m_\chi^2/m_N$, which drops rapidly with DM mass. This corresponds to the right panel of Fig. 1, where the resonant response only appears in a corner of the allowed phase space. The maximum recoil energy occurs in the "hard-wall" limit where the DM bounces off the heavy nucleus.

In the above discussion, we have considered a fixed DM velocity $v$ with typical $v \sim 10^{-3}$. Larger recoil energies are possible from DM with greater speeds, but the likelihood of an incident DM with larger velocities eventually becomes exponentially suppressed and it is typically assumed there is essentially no DM above the local escape velocity, which translates to $v \lesssim 3 \times 10^{-3}$ in the lab frame. Eq. (35) implies that for a given $\omega$, $q$ in the scattering phase space, there is a minimum DM initial velocity required:

$$v_{\min}(q, \omega) = \frac{\omega_{\mathbf{q}}}{q} + \frac{q}{2m_\chi}. \tag{39}$$

For nuclear recoils, this becomes $v_{\min} = \sqrt{m_N E_R/2\mu_{\chi N}^2}$. We can see this restriction explicitly in the rate by taking an isotropic approximation, in which we assume the target-dependent piece of Eq. 35 depends only on $q$ and not on $\mathbf{q}$. (Including the full $\mathbf{q}$ dependence can be important, however, for anisotropic target systems.) Using the delta function $\delta(\omega - \omega_{\mathbf{q}})$ to integrate Eq. (24) over the angle $\hat{\mathbf{q}} \cdot \hat{\mathbf{v}}$, we obtain the isotropic rate:

$$R_\chi^{\mathrm{iso}} = \frac{1}{\rho_T} \frac{\rho_\chi}{m_\chi} \int \frac{q\,dq}{(2\pi)^2} \, d\omega \, \eta(v_{\min}(q, \omega)) \times \frac{\pi \bar{\sigma}(q)}{\mu_\chi^2} \times S(q, \omega) \,, \tag{40}$$

where we have introduced a function for the mean inverse DM speed:

$$\eta(v_{\min}) = \int_{v > v_{\min}}^{\infty} d^3\mathbf{v} \, \frac{f_\chi(\mathbf{v})}{v}. \tag{41}$$

Anisotropic effects are an important feature of scattering in condensed matter targets, but for simplicity we will work in the isotropic approximation and use Eq. (40) throughout.

★ *Exercise:* Take the heavy mediator limit of the interaction in Eq. (33), and show that it gives the dynamic structure factor given in Eq. (36). Using Eq. (24), show that this reproduces the usual WIMP scattering rate. To account for the nuclear recoil form factor, use the fact that the initial and final states are many-body nucleon states, and for elastic recoils the initial and final states are the same (up to an overall boost given by the final recoil of the nucleus).

## 2.4 Direct detection with bound systems

The previous section showed that nuclear recoils are not well matched to the allowed scattering kinematics of sub-GeV dark matter. What we seek, instead, are systems with strong response in the relevant $q, \omega$ for keV-GeV dark matter. This is a quite broad range of DM masses, of course, and different systems will be useful throughout this mass range. Rather than being limited by the free-particle kinematics assumed for nuclear recoils, the response of bound systems (including atomic, molecular, and condensed matter systems) is important to consider for two reasons. The first is that any realistic target system must be treated as a bound system at low energies and momentum transfers, which will overlap with the regime relevant for sufficiently low mass DM. The second is that the bound system response can differ significantly from free particle response, and in particular condensed matter systems offer an enormous array of possible excitations and response functions (dynamic structure factors) to which the DM can couple. Fig. 2 illustrates these ideas with possible excitations in various bound systems.

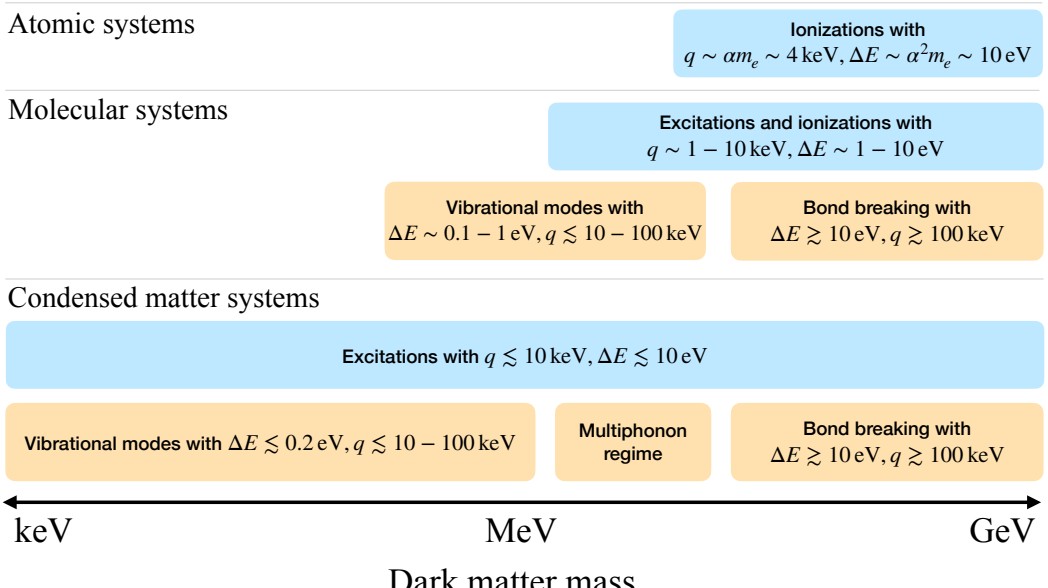

Figure 2: For various bound systems, we indicate some possible excitation modes in the system, some typical $(q, \omega \sim \Delta E)$ where they give a large response, and the corresponding range of DM masses which are well-matched to that system. Blue shaded boxes indicate electronic excitations, while orange shaded boxes indicate modes that can be excited by a DM-nucleus coupling. For electronic excitations in condensed matter systems, different materials will have ideal response to different ranges of DM masses, so the shaded region just indicates that there exist known systems that have been studied for the whole mass range. For these lectures, we will focus on spin-independent DM interactions with electrons and/or nuclei. For spin-dependent interactions, other excitation modes and response functions can be considered.

The general dynamic structure factor we will consider is given by:

$$S(\mathbf{q}, \omega) \equiv \frac{2\pi}{V} \sum_f |\langle f | \left( \sum_k f_e e^{i\mathbf{q}\cdot\mathbf{r}_k} + \sum_I f_I e^{i\mathbf{q}\cdot\mathbf{r}_I} \right) |i\rangle|^2 \delta(E_f - E_i - \omega), \qquad (42)$$

where $f_e$ is a (normalized) DM coupling with electrons and $f_I$ is the DM coupling to ions, and we have summed over all constituents of the system. Note that different conventions exist in the literature for the overall normalization of $S(\mathbf{q}, \omega)$ in terms of factors of $2\pi$ and volume, and the couplings $f_{e,I}$ are typically normalized relative to an overall interaction strength. For example, for the dark photon model, we have the natural definition $f_e = -1$ and $f_I = Z_I$, while the strength of the interaction potential is absorbed in $\bar{\sigma}(q)$. Note also that we will focus on the particular choice of structure factor above, which depends only on the position operators for electrons $\mathbf{r}_k$ and ions $\mathbf{r}_I$, as this is the structure factor relevant for the most commonly-studied models in the literature. In other models, the leading nonrelativistic coupling could have additional dependence on the target momenta and spins, and requires defining additional structure factors.

In Eq. (42), the initial and final states $|i\rangle, |f\rangle$ are in general *many-body* states of the bound system. A key quantity which determines the importance of including these many-body states is the momentum transfer $\mathbf{q}$ from the DM to the target system. For sufficiently large $\mathbf{q}$, the relative phases between the electrons or nuclei will average out to zero, and the scattering can

be effectively treated as if DM interacted incoherently with an individual electron or nucleus. However, when $q$ becomes comparable to the momentum spread of the bound wavefunctions, we must consider the bound states. In the remainder of this section, we will give an overview of the dynamic structure factor with various systems and excitations, and how this can guide our intuition in the search for optimal target systems for dark matter direct detection.

# 3   Electron excitations

The phenomenology of DM-electron scattering is dominated by the fact that electrons are bound in atomic, molecular, and solid-state systems, with wavefunctions that are very far from momentum-eigenstate plane waves. The typical length or momentum scale for electronic wavefunctions is set by the Bohr radius:

$$a_0 = \frac{1}{\alpha m_e} = 5.29 \times 10^{-11} \text{ m}, \qquad p_0 \equiv \frac{1}{a_0} = 3.73 \text{ keV}. \tag{43}$$

In the ground state of the hydrogen atom, $\langle r \rangle = \frac{3}{2} a_0$; in an larger atom, the larger value of the principal quantum number $n$ is partially compensated by the increased screened nuclear charge, giving a parametrically similar answer. In a molecule, the interatomic distance is set by minimizing the total energy of covalently-bonded atoms, and since the atomic wavefunctions must overlap to bond, the bond length is also of order $a_0$; for example, the carbon-carbon bond length in organic molecules is 0.14 nm $\simeq 2a_0$. The same logic holds for solid-state lattices (silicon has a minimum interatomic distance of $\sim 4.4a_0$ and a lattice constant of $\sim 10a_0$), and even the Fermi momentum $k_F$ for delocalized electrons in a metal is of order $p_0$, since it depends on the number density of electrons and hence is set in part by the lattice spacing. The ground state position-space orbitals are localized as $\exp(-r/a_0)$, yielding momentum-space orbitals which fall off at large $p$ as a power law. The energies of electronic states are parametrically set by the Rydberg energy: 13.6 eV for the ionization energy of hydrogen, $\mathcal{O}(10)$ eV for outer-shell binding energies in noble atoms, $\mathcal{O}(5)$ eV for excitation gaps in organic molecules, and $\mathcal{O}(1-5)$ eV for semiconductor gaps. As shown in Fig. 2, these binding energies decrease from isolated atoms to molecules to solid-state systems.

We can compare these scales to the typical energy and momentum scales of DM. The minimum and maximum momentum transfer required to create an excitation $\omega$ can be obtained by setting $\mathbf{q} \cdot \mathbf{v} = qv$ in Eq. (35) and solving for $q$. The minimum is given by

$$\begin{aligned} q_{\min} &= m_\chi v - \sqrt{(m_\chi v)^2 - 2m_\chi \omega} \\ &\simeq \frac{\omega}{v}, \end{aligned} \tag{44}$$

where in the second line we have taken the large $m_\chi$ limit (for smaller $m_\chi$, $q_{\min}$ is strictly larger). Taking $v = v_{\max} = v_\oplus + v_{\text{esc}} \simeq 800$ km/s $= 2.67 \times 10^{-3}$, we find

$$q_{\min} \simeq p_0 \left( \frac{\omega}{10 \text{ eV}} \right). \tag{45}$$

For the maximum momentum transfer, we similarly find

$$\begin{aligned} q_{\max} &= m_\chi v + \sqrt{(m_\chi v)^2 - 2m_\chi \omega} \\ &\simeq 2m_\chi v \simeq p_0(m_\chi/0.7\,\text{MeV}). \end{aligned} \tag{46}$$

From this, we learn a number of things:

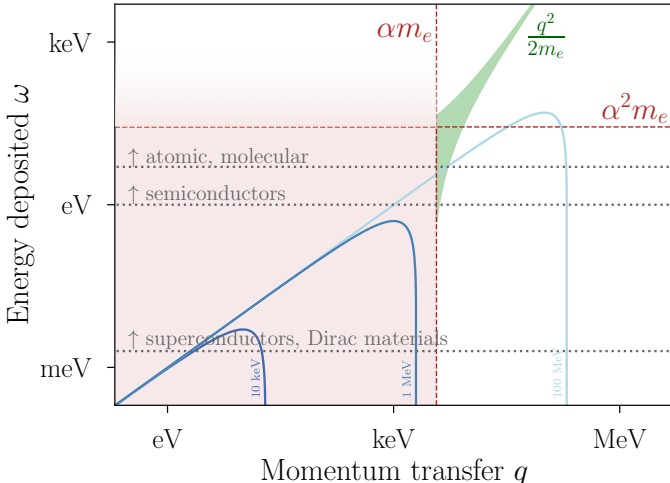

Figure 3: For DM-electron interactions, the response at high $q$ is peaked about the free-electron dispersion $\omega = q^2/(2m_e)$. The dashed lines at $q \sim 1/a_0 = \alpha m_e$ and $\omega \sim \alpha^2 m_e$ indicate typical scales for the wavefunction spread and energies of bound electrons. Many-body effects are expected to be particularly important at lower $q, \omega$, indicated by the shaded region. Depending on the target material and detection method, the relevant response function will be cut off at low $\omega$; we show typical gaps for ionization in atomic systems, scintillation in molecules, electron-hole excitations in semiconductors, and gapped excitations in superconductors and Dirac materials. Kinematically allowed regions for DM scattering are shown for $m_\chi = 10$ keV, 1 MeV, and 100 MeV at $v = 10^{-3}$, as in Fig. 1.

- MeV–GeV DM has the correct kinematics to access the electronic response for atomic and molecular systems where they have strong support, as shown schematically in Fig. 3. However, it can also be seen from Fig. 3 that the DM scattering kinematics is not necessarily ideally matched to the response. For example, for atomic ionization, the rate will be strongly peaked at low ionization energies, since larger $\omega$ requires accessing the high-momentum tail of the electron wavefunctions which is power-law suppressed. Thus, while it is kinematically permitted for (say) 100 MeV DM to deposit *all* of its $\sim 50$ eV of kinetic energy on an atomic electron, it is extremely unlikely to do so. Similarly, this will favor scattering on the high-velocity tail of the DM velocity distribution, which implies a large increase in rate as the gap is lowered [32].

- For conventional semiconductors with $\mathcal{O}(\text{eV})$ gaps, DM *necessarily* probes distance scales smaller than the lattice constant. Taking silicon as an example, with $\omega = 1$ eV, we find from Eq. (45) $q_{\min} \simeq p_0/10 \simeq (10a_0)^{-1}$, which is the inverse lattice spacing. Thus, while it is true that the valence electrons are delocalized across the lattice, the particular kinematics of DM scattering weights the the quasi-localized portion of the electronic wavefunctions. Many-body effects will therefore be less important for conventional semiconductors than for lower-gap materials.

- Accessing the true long-range behavior of delocalized electrons *requires* a narrow-gap material, many of which have rather exotic electronic properties. Since these narrow gaps are mandatory to probe sub-MeV DM which carries sub-eV kinetic energies, the search for novel materials with the required electronic properties, involving close collaboration with condensed matter physicists, is a key component of the active research

in light DM detection.

## 3.1 Atomic systems

The first application of dark matter-electron scattering involved electron ionization in liquid xenon [33,34], but to illustrate the essential features we will deal with a simpler toy example, dark matter scattering off a single hydrogen atom [1,35]. Throughout this section, we will take the benchmark dark photon model, where DM couples to both protons and electrons with a spin-independent potential. Summing over all target atoms (or nuclei), the dynamic structure factor is

$$S(\mathbf{q},\omega) = \frac{2\pi N_{\text{nuc}}}{V} \sum_f |\langle f | e^{i\mathbf{q}\cdot\mathbf{r}_N} - e^{i\mathbf{q}\cdot\mathbf{r}_e} | 0 \rangle|^2 \delta(E_f - E_0 - \omega)\,, \tag{47}$$

where $\mathbf{r}_N$ and $\mathbf{r}_e$ are the nuclear and electronic coordinates, respectively, and $|f\rangle$ represents an excited electronic state of the atom. In the approximation where the nucleus is infinitely heavy and thus stationary, we may set $\mathbf{r}_N = 0$.

The excited electron state $|f\rangle$ may be either a bound state or a continuum state, the latter of which corresponds to an ionized electron. Since the formalism of atomic electron scattering is typically applied to liquid noble element detectors which are sensitive to ionized electrons, we will focus on the continuum states. The initial state is simply the ground state of the hydrogen atom, $\psi_{100}(\mathbf{r}) = 2a_0^{-3/2} e^{-r/a_0}$. Excited states are labeled by a wavevector $\mathbf{k}$, so the sum over $|f\rangle$ turns into an integral, and the structure factor becomes

$$S(\mathbf{q},\omega) = \frac{2\pi N_{\text{nuc}}}{V} \int \frac{d^3\mathbf{k}}{(2\pi)^3} \delta(E_{\mathbf{k}} - E_0 - \omega) |f_{0\to\mathbf{k}}(\mathbf{q})|^2\,, \tag{48}$$

where

$$f_{0\to\mathbf{k}}(\mathbf{q}) = \int d^3\mathbf{r}\, \psi_{\mathbf{k}}^*(\mathbf{r})\psi_{100}(\mathbf{r})e^{i\mathbf{q}\cdot\mathbf{r}} \tag{49}$$

is the *atomic form factor* for transitions between the ground state and the continuum state $\mathbf{k}$. Asymptotically far away from the nucleus, the final-state electron behaves as a free particle, so we have absorbed an extra factor of $\sqrt{V}$ inside $\psi_{\mathbf{k}}^*(\mathbf{r})$, such that it behaves as $\psi_{\mathbf{k}}^*(\mathbf{r}) \propto e^{i\mathbf{k}\cdot\mathbf{r}}$ as $r \to \infty$. Similarly, we may define $\mathbf{k}$ through the energy of the continuum state as $E_{\mathbf{k}} = \frac{k^2}{2m_e}$. Typically we are interested in the energy spectrum of ionized electrons $dR/dE_{\text{er}}$, so using $dE_{\text{er}} = dE_{\mathbf{k}} = k\,dk/m_e$, so we may trade the integral over $k$ for an integral over $E_{\text{er}}$. Collecting the various normalization factors, and decomposing the outgoing wavefunction into spherical waves with angular quantum numbers $l'$ and $m'$, it is convenient to define an *ionization form factor*,

$$|f_{\text{ion}}(k,q)|^2 = \sum_{l',m'} \frac{2k^3}{(2\pi)^3} |f_{0\to k,l',m'}(\mathbf{q})|^2\,, \tag{50}$$

where the factor of 2 accounts for spin degeneracy. The radial part $\widetilde{R}_{kl}$ of the ionized wavefunctions is normalized as

$$\int dr\, r^2 \widetilde{R}_{kl}^*(r)\widetilde{R}_{k'l'}(r) = (2\pi)^3 \frac{1}{k^2}\delta_{ll'}\delta(k-k')\,, \tag{51}$$

so that $\widetilde{R}_{kl}(r)$ itself is dimensionless, and therefore so is $f_{\text{ion}}$. Using Eq. 40 since the target system is spherically symmetric, we obtain the total differential rate per unit detector mass,

$$\frac{dR}{d\ln E_{\text{er}}} = N_T \frac{\rho_\chi}{m_\chi} \frac{\bar{\sigma}_e}{8\mu_{\chi e}^2} \int dq\, q\, |F_{\text{DM}}(q)|^2 |f_{\text{ion}}(k,q)|^2 \eta(v_{\text{min}})\,, \tag{52}$$

where for electron scattering

$$v_{\min} = \frac{E_{\mathrm{er}} + |E_0|}{q} + \frac{q}{2m_\chi} \tag{53}$$

and the form factor for a dark photon mediator $F_{\mathrm{DM}}(q)$ was defined in Eq. 30. $N_T$ is the number of target nuclei per unit detector mass, and $E_0 = -13.6$ eV is the binding energy of hydrogen.

For the Hydrogen atom, the form factors can be computed exactly. In particular, for a nucleus with charge $Z$, the exact outgoing radial wavefunctions are [36]

$$\widetilde{R}_{kl}(r) = (2\pi)^{3/2} \frac{\sqrt{\frac{2}{\pi}} \left| \Gamma\left(l+1+\frac{iZ}{ka_0}\right) \right| e^{\frac{\pi Z}{2ka_0}}}{(2l+1)!} e^{ikr} \, {}_1F_1\left(l+1+\frac{iZ}{ka_0}, 2l+2, 2ikr\right), \tag{54}$$

where $\Gamma$ is the gamma function, ${}_1F_1$ is the confluent hypergeometric function, and we can take $Z = 1$ arbitrary to facilitate later comparison with larger hydrogenic atoms. Since this is an eigenstate of the same potential which determines the ground state, it is automatically orthogonal to $\psi_{100}$ when $Z = 1$. Using this for the outgoing states, the ionization form factor can be obtained as [35]

$$|f_{\mathrm{ion}}(k,q)|^2 = \frac{512 Z^6 k^2 q^2 a_0^4 ((3q^2+k^2)a_0^2 + Z^2) \exp\left[-\frac{2Z}{ka_0} \tan^{-1}\left(\frac{2Zka_0}{(q^2-k^2)a_0^2+Z^2}\right)\right]}{3((q+k)^2 a_0^2 + Z^2)^3 ((q-k)^2 a_0^2 + Z^2)^3 (1 - e^{-\frac{2\pi Z}{ka_0}})}. \tag{55}$$

Note that this vanishes as $q \to 0$, as required by orthogonality of the initial and final states. The dynamic structure factor for ionization of atomic hydrogen is then given by

$$S(\mathbf{q}, \omega) = \pi n_{\mathrm{atom}} \frac{m_e}{k^2} \times |f_{\mathrm{ion}}(k,q)|^2, \quad k = \sqrt{2m_e(\omega - E_0)}. \tag{56}$$

We plot this in Fig. 4, as compared with the kinematic restriction for DM scattering, which requires us to be to the right of the dashed line. As discussed in the introduction to this section, this implies that we obtain the largest rates for DM on the tail of the velocity distribution and favoring low $\omega$. Furthermore, it is clear that atomic hydrogen is not an ideal direct detection target given that the region where the dynamic structure factor is largest is not entirely accessible to DM.

With $k \sim q \sim \alpha m_e$ and $n_{\mathrm{atom}} \sim (\alpha m_e/10)^3$, we can estimate $|S(\mathbf{q}, \omega)| \sim (\alpha m_e)^2/\alpha$. Using this in Eq. (40) with $\omega \sim \alpha^2 m_e$ gives an order of magnitude estimate for the rate when $m_\chi \gg m_e$:

$$R_\chi \sim \frac{1}{\rho_T} \frac{\rho_\chi}{m_\chi} \bar{\sigma}_e \frac{\alpha^2}{v} (\alpha m_e)^3. \tag{57}$$

For $m_\chi = 10$ MeV and $\overline{\sigma}_e = 10^{-37}$ cm$^2$ at the freezeout cross section, this gives an expected rate of $\mathcal{O}(10)$ events/s/kg, an enormous rate compared to WIMP experiments. However, this estimate is of course an overestimate, since we are using typical values of $S(q, \omega)$ closer to the peak. The kinematic restriction to the allowed DM phase space shown in Fig. 4 implies we only access the structure factor when it is smaller, which can give a suppression by a few orders of magnitude.

For a general atom, it is expected that the qualitative behavior is quite similar for the outermost electrons. However, one no longer has exact solutions for the wavefunctions and the results for the structure factor are more uncertain. One approach is to use the Hartree-Fock approximation to construct the approximate many-electron bound-state wavefunctions as a Slater determinant of single-particle orbitals. One can then compute the scattering rate using single-particle orbitals, with many-body effects included in the chosen form of the orbitals,

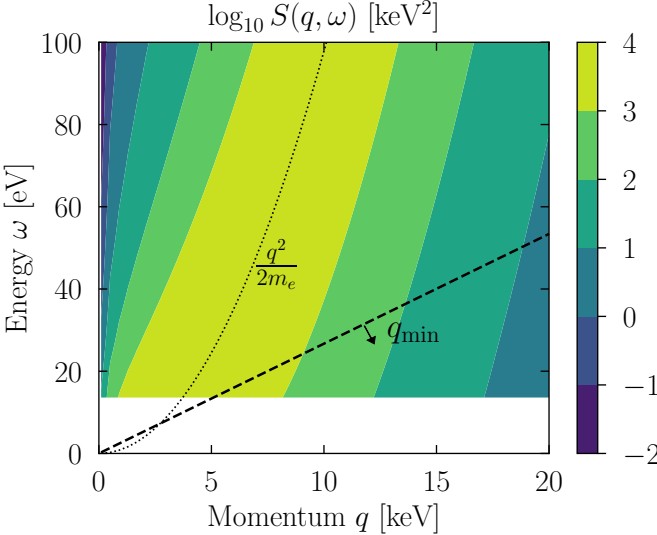

Figure 4: The dynamic structure factor for ionization of hydrogen, obtained using Eq. (55) with an arbitrary reference $n_{\text{atom}} = 1/(a_0)^3$. For $q \gg p_0$, the bound nature of the electron becomes less important and the peak of the structure factor converges to the free-particle dispersion $\omega = q^2/(2m_e)$, indicated by the dotted line. The dashed line is the minimum $q$ for DM scattering, Eq. (45).

which are tabulated in the literature. In the Hartree-Fock approximation, the radial wavefunctions for each orbital can be expressed in a basis of Slater-type orbitals with effective charges $Z_{jl}$ and coefficients $C_{jln}$ as [37]

$$R_{nl}(r) = a_0^{-3/2} \sum_j C_{jln} \frac{(2Z_{jl})^{n'_{jl}+1/2}}{\sqrt{(2n'_{jl})!}} \left(\frac{r}{a_0}\right)^{n'_{jl}-1} e^{-Z_{jl}r/a_0}. \tag{58}$$

These wavefunctions are known as Roothaan-Hartree-Fock (RHF) wavefunctions after a standard technique in quantum chemistry for solving the Hartree-Fock equations. However, Ref. [37] does not provide parameterizations of the continuum wavefunctions, the earlier applications of this formalism [34, 38] had to supply the final-state wavefunctions externally, which were not guaranteed to be orthogonal to the bound states and thus neglect many-body effects in a possibly important way. Indeed, in the earliest literature [33], a plane-wave approximation was used for computational simplicity, along with adding in a Fermi factor by hand.

As the atomic number of an atom increases, relativistic effects also become more important. These can be incorporated [39, 40] with the Dirac-Hartree-Fock approximation, for which a public code, FAC, exists to calculate both the bound and continuum wavefunctions from a self-consistent potential [41], ensuring orthogonality. The combination of relativistic and many-body effects (which are expected to be important for excited states) in xenon produces a spectrum which differs by almost an order of magnitude at both small and large ionization energies, suppressing the rate at small energies but drastically increasing the tail at large energies, which can potentially have a large impact on experimental searches [40]. As of yet there is no direct measurement of the ionization form factors of xenon and argon in the relevant kinematic regime for sub-GeV DM-electron scattering, and thus each of the above approximations for the wavefunctions must be considered to carry some unquantified systematic uncertainty.

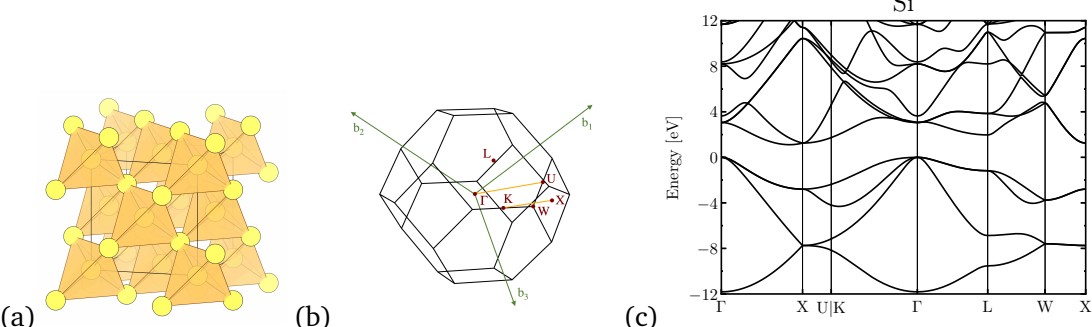

Figure 5: (a) Real-space face-centered cubic (FCC) lattice structure for Si (b) First Brillouin Zone for FCC crystal (c) Electron band structure for Si. The reciprocal space primitive vectors are given by $\mathbf{b}_1 = 2\pi/a(-1, 1, 1)$, $\mathbf{b}_2 = 2\pi/a(1, -1, 1)$, and $\mathbf{b}_3 = 2\pi/a(1, 1, -1)$. $\Gamma$ labels the origin of the reciprocal space while the capital Roman letters indicate standard high-symmetry points in the BZ. Graphics reproduced from Ref. [42], which also gives examples of additional crystals and target materials.

## 3.2 Solid state systems

To probe dark matter with MeV mass or below, electronic excitation energies at the eV scale and below are required, necessitating the use of solid-state systems with the required low band gaps. Historically, much of the theoretical and experimental effort has focused on *conventional semiconductor* detectors – specifically silicon and germanium – as well as *conventional superconductors* like aluminum. Recently there has been a flourishing effort to identify new materials with particular properties which are well-suited to the kinematics of sub-GeV DM.

The difficulties with writing down the true many-body states of the crystal are even worse than for atoms, but fortunately an effective single-particle description is possible. One of the primary tools in condensed matter is that of Density Functional Theory (DFT), which provides a first principles approach to determine single-particle wavefunctions self-consistently in terms of effective potentials due to all the other electrons. For electrons in a periodic potential satisfying $V(\mathbf{r}) = V(\mathbf{r}+\mathbf{R})$, the single-particle wavefunctions are called Bloch states, and given by

$$\psi_\mathbf{k}(\mathbf{r}) = \frac{1}{\sqrt{V}}e^{i\mathbf{k}\cdot\mathbf{r}}u_\mathbf{k}(\mathbf{x}), \qquad u_\mathbf{k}(\mathbf{r}+\mathbf{R}) = u_\mathbf{k}(\mathbf{r}), \tag{59}$$

where $V$ is the total volume of the crystal and $u_\mathbf{k}$ is a cell function which is also periodic. $\mathbf{R}$ is a lattice vector which is any integer linear combination of three basis vectors called the primitive lattice vectors, $\mathbf{a}_1, \mathbf{a}_2, \mathbf{a}_3$. The above result is known as *Bloch's theorem*, and has the important implication that *valence electrons are delocalized*: their wavefunctions have support throughout the entire crystal, thanks to the constant modulus of the phase factor and the periodicity of the cell function. Solving the single-particle Schrödinger equation with the ansatz (59) for the wavefunction will yield a discrete set of quantized energy eigenvalues and eigenfunctions for each $\mathbf{k}$, which may be labeled with an integer $n$, called the *band index*, and restricting to inequivalent solutions restricts $\mathbf{k}$ to a region of the reciprocal lattice space known as the *first Brillouin zone (BZ)*. A diagram of such a band structure, along with the associated real-space crystal lattice and BZ diagram is shown in Fig. 5. The vector $\mathbf{k}$ is called a *crystal momentum*, and differs from the physical momentum because crystal momentum is only conserved up to the addition of an arbitrary reciprocal lattice vector $\mathbf{G}$ where $\mathbf{G}\cdot\mathbf{R} = 1$.

The first calculations of DM-electron scattering in silicon and germanium used either DFT band structures and wavefunctions [33, 43] or semi-analytic models based on hydrogenic or

tight-binding orbitals [32, 44]. By the arguments at the beginning of this section, the typical momentum transfers probe electrons on length scales smaller than a single unit cell as long as the gap is $\mathcal{O}(\text{eV})$ or larger, so both of these approaches are expected to give the correct order of magnitude for the total scattering rate, though there are important differences between the spectra at small and large recoil energies. The original DFT approach of Ref. [43] amounts to the following procedure: start with Eq. (42) assuming only electron interactions, treat the initial and final states as Bloch states, and neglect the sum over all electrons in the operator by taking $\sum_k e^{i\mathbf{q}\cdot\mathbf{r}_k} \to e^{i\mathbf{q}\cdot\mathbf{r}}$. Then the dynamic structure factor can be written as

$$S(\mathbf{q}, \omega) = \frac{2\pi}{V} \sum_{\mathbf{k}, \mathbf{k}', i, i'} |\langle \mathbf{k}', i'|e^{i\mathbf{q}\cdot\mathbf{r}}|\mathbf{k}, i\rangle|^2 \delta(\omega + E_{i\mathbf{k}} - E_{i'\mathbf{k}'}), \tag{60}$$

where $i$ denotes occupied valence state, $i'$ denotes an unoccupied conduction state (at zero temperature), and $|\mathbf{k}, i\rangle$ is a Bloch state. However, the assumption that we can use single-particle states for the initial and final excitations is not correct in general.

More recently, Refs. [45, 46] showed instead that the true dynamic structure factor in terms of many-body states is given instead in terms of the dielectric function $\epsilon(\mathbf{q}, \omega)$:

$$S(\mathbf{q}, \omega) = \frac{q^2}{2\pi\alpha} \text{Im}\left(-\frac{1}{\epsilon(\mathbf{q}, \omega)}\right) \tag{61}$$

$$= \frac{2\pi}{V|\epsilon(\mathbf{q}, \omega)|^2} \sum_{\mathbf{k}, \mathbf{k}', i, i'} |\langle \mathbf{k}', i'|e^{i\mathbf{q}\cdot\mathbf{r}}|\mathbf{k}, i\rangle|^2 \delta(\omega + E_{i\mathbf{k}} - E_{i'\mathbf{k}'}). \tag{62}$$

This applies to both scalar and vector mediators coupling to electrons (including dark photons), since the leading non-relativistic coupling in all these cases is just a spin-independent Yukawa potential between DM and electrons. In the equation above, the second line can be shown explicitly by using the Lindhard formula for the dielectric function [45], which is essentially an approximation to the dielectric function in a noninteracting gas, *i.e.* computing the leading order polarization function in an electron gas. Now an additional factor of $1/|\epsilon(\mathbf{q}, \omega)|^2$ appears compared to Eq. (60). The difference arises from the fact that the true initial and final states are many-body states, and the actual result in terms of single-particle states must account for the total response of the medium, which screens any external perturbation by $\epsilon(\mathbf{q}, \omega)$. In other words, the factor of $1/|\epsilon(\mathbf{q}, \omega)|^2$ accounts for in-medium screening effects and is equivalent to resumming an infinite series of insertions of the polarization loop, where we see explicitly that the structure factor includes terms to all orders in $\alpha$.

The result from including the full many-body response with screening leads to qualitatively different behavior in the low $q$ limit. For example, in generic solid-state systems (including both semiconductors like silicon and metals like aluminum), there is a resonance for $q \lesssim p_F$ called the *plasmon*, which appears in the dynamic structure factor as

$$S(\mathbf{q}, \omega) \propto q^2 \omega \frac{\omega_p^2 \Gamma_p}{(\omega_p^2 - \omega^2)^2 + \omega^2 \Gamma_p^2}, \tag{63}$$

with $\Gamma_p$ a finite width which regulates the resonance. The appearance of the *plasma frequency*

$$\omega_p = \sqrt{\frac{4\pi\alpha n_e}{m_e}} \tag{64}$$

suggests an interpretation of this resonance as the collective oscillation of the entire valence electron density $n_e$, which is not visible in a picture of single-particle wavefunctions. The quantized mode corresponding to the collective excitation is also known as a plasmon, where

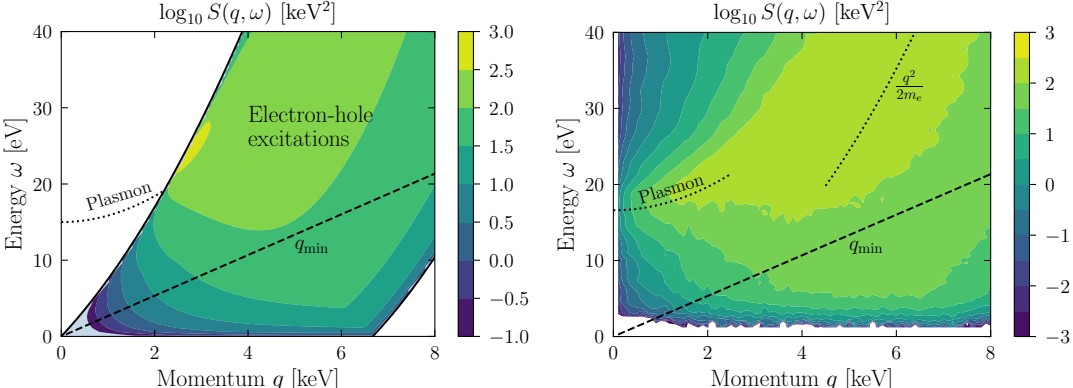

Figure 6: (*left*) Dynamic structure factor for a free electron gas with $\omega_p = 15$ eV and $k_F = 3.3$ keV. The plasmon resonance shows up as an infinitely narrow resonance indicated by the dotted line, while the rest of the support is interpreted as electron-hole excitations. (*right*) Dynamic structure factor for ionization in a Si semiconductor, based on the calculation in Ref. [45]. At low $q$, it is peaked at the plasmon resonance, while at high $q$ the peak converges to the free-electron dispersion $q^2/(2m_e)$, similar to the structure factor for Hydrogen. In both panels, the dashed line is the minimum $q$ for DM scattering, Eq. (45), showing that the peaks of the structure factor are not accessible to halo DM.

we can interpret this result as this structure factor for producing a single plasmon. This mode can be seen most easily in calculating the response of a degenerate electron gas, which gives an infinitely narrow resonance where $\epsilon(\mathbf{q}, \omega) = 0$, as illustrated in Fig. 6 (left panel). This response function was calculated assuming a spherical Fermi surface, leading to a restricted part of phase space for electron-hole excitations. In a realistic material, electron-hole excitations are no longer so restricted and the plasmon can decay to electron-hole excitations. A DFT-based calculation for Si is shown in the right panel of Fig. 6.

From Eq. (61), we can estimate the size of the response to be $|S(\mathbf{q}, \omega)| \sim q^2/(2\pi\alpha) \sim 100$ keV$^2$, which is borne out by the calculations shown in Fig. 6. Using $|S(\mathbf{q}, \omega)| \sim q^2/(2\pi\alpha)$ in Eq. (40) with $q \sim \alpha m_e$ and $\omega \sim \alpha^2 m_e$ gives an order of magnitude estimate for the rate when $m_\chi \gg m_e$:

$$R_\chi \sim \frac{1}{\rho_T} \frac{\rho_\chi}{m_\chi} \bar{\sigma}_e \frac{\alpha^2}{v} (\alpha m_e)^3 \,, \tag{65}$$

which is similar to the estimate we obtained from the atomic dynamic structure factor, Eq. (56). However, similar to the atomic case, we see from Fig. 6 that the peaks of the dynamic structure factor are again not kinematically matched to the allowed DM scattering phase space, which requires $\omega < q v_{max}$. At $q > k_F$, the peak at $\omega \sim q^2/(2m_e)$ is not accessible, since in this regime $\omega \gtrsim q k_F/m_e \sim q\alpha$. At low $q < k_F$, most of the weight in the ELF is carried by the plasmon which is located at $\omega > q v_F$. However, the DM velocity is typically much slower than the Fermi velocity in conventional materials, so that the DM cannot access the plasmon. However, semiconductors are still better than a typical atomic target as in Fig. 4, due to the lower gaps. This motivates the search for more ideal targets to search for DM-electron interactions, although to date Si and Ge still remain among the best targets for DM masses above MeV. For sub-MeV dark matter, the behavior of the dynamic structure factor in the $q \lesssim$ keV and $\omega <$ eV regime becomes important. In this mass range, some of the materials proposed include that of superconductors [46, 52] and Dirac materials [46, 49, 53, 54], which we will not discuss in detail here. Cross section curves for these materials are included for reference in Fig. 7.

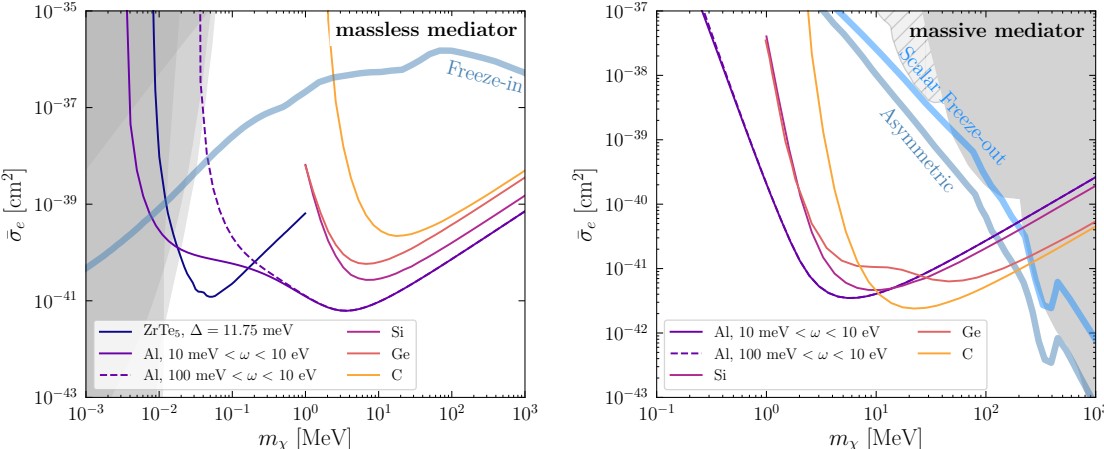

Figure 7: Reach to DM-electron cross section in different solid state targets for scattering through a light mediator (left) and heavy mediator (right), assuming kg-year exposure and zero background. Diamond (C) gives an example of a somewhat high gap target $E_{\text{gap}} = 5.5$ eV; the reach shown is from Ref. [47]. Si and Ge have O(eV) gaps and have been studied extensively as target materials. The reach for Ge with a massless mediator and Si in both cases is from Ref. [45]. For Ge and a massive mediator, the reach shown is from Ref. [48]; for DM masses above ∼ 20 MeV, the reach is dominated by the excitation of semi-core electrons. For sub-MeV DM, lower gap materials are needed and we show projections for an example Dirac material from Ref. [49] and for Al from Refs. [45,46]. In the left plot, the thick blue line is the predicted cross section if all of the relic DM is produced by freeze-in interactions [9,33] and the shaded regions are constraints from stellar emission [8, 50]. In the right plot, the thick blue lines are cross sections for freezeout of scalar DM or fermionic asymmetric DM [38]; note that for asymmetric DM, the line is a lower bound and cross sections above it also satisfy relic density and CMB annihilation considerations. The shaded region shows combined direct detection bounds (solid grey) and model-dependent accelerator bounds when the dark photon mass is $m_{A'} = 3m_\chi$ (hatched grey) [51]. All bounds and relic density lines assume a dark photon mediator.

# 4  Phonon excitations

Similar to the case of electron excitations, at low energies we must account for the fact that nuclei are bound in atomic, molecular, and solid-state systems. As the DM mass drops below ∼ GeV, the free-nucleus recoil energy ∼ $(m_\chi v)^2/m_N$ becomes comparable to the typical energy to break a molecular bond or displace an ion in a crystal, $\mathcal{O}(10)$ eV. Indeed, a possible direct detection signature of sub-GeV might be chemical bond breaking or production of defects in crystals [55,56]. In the DM mass range above $\mathcal{O}(10)$ MeV where bond breaking is possible, the typical momentum transfer $q \sim m_\chi v \gtrsim \mathcal{O}(10)$ keV. These momentum transfers are sufficiently large that it is still possible to treat the scattering as occurring off individual bound nuclei in a crystal, for instance. However, the bound state nature induces some spread in the response compared to the resonant free elastic recoil in Eq. (36), which can be also interpreted as a multi-phonon response of the system [51, 57]. For DM mass below ∼ 1 MeV, the maximum momentum transfer is $q < 1$ keV, comparable to the inverse interparticle spacing. Then we must account for the fact that the nuclei (ions) are all coupled, the fundamental modes are

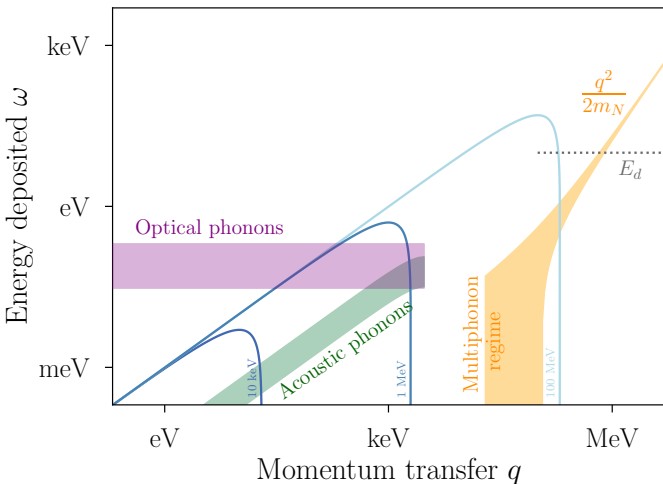

Figure 8: For DM-nucleus interactions, the response at high $q$ is highly peaked about the free-nucleus dispersion $\omega = q^2/(2m_N)$. As the energy drops below the displacement energy for a nucleus in a potential, $E_d$, the response about the free-nucleus dispersion broadens and we enter the multiphonon regime at $q \sim 10 - 100$ keV. For $q$ well below $\sim 10$ keV, the dynamic structure is dominated by resonant response on the acoustic and optical phonon dispersions, corresponding to single phonon excitations. Kinematically allowed regions for DM scattering are shown for $m_\chi = 10$ keV, 1 MeV, and 100 MeV at $v = 10^{-3}$.

vibrational modes called phonons, and DM can scattering coherently off the crystal.[5] This leads to single and few phonon excitations at low energies. The various regimes of nuclear response are illustrated in Fig. 8, as compared with the kinematically allowed regions for DM of various masses. Phonon excitations are particularly interesting since they naturally offer a more optimal (and resonant) response in the sub-MeV DM scattering phase space as compared with existing systems identified for electron scattering. In this section, we give a brief introduction to phonon excitations in crystals to illustrate this point.[6].

## 4.1 Introduction to phonons

Introductions to phonon modes can be found in standard references [59], and for completeness we provide a brief review. The system we will consider is a 1D regular lattice of $N$ atoms of mass $M$, shown in the top left of Fig. 9. All of the atoms are identical, so the unit cell has a size $a$ (the lattice spacing) and contains one atom. Each atom $i$ has a possible displacement from its equilibrium position, denoted by $u_i$. The Hamiltonian for this system is modeled with an effective potential for the relative displacements of neighboring atoms:

$$H = \sum_i \frac{1}{2} M \dot{u}_i^2 + \frac{1}{2} k_{\text{eff}} (u_{i+1} - u_i)^2 + \dots ,  \tag{66}$$

---

[5]For a given atom, we can think of the nucleus plus the most tightly bound electrons as being relatively unaffected by the presence of the other atoms. Meanwhile, the outer shell electrons of the atom interact with electrons of neighboring atoms, giving rise to the delocalized electron wavefunctions and complex electron band structure in a material. Therefore, in what follows we will interchangeably use nucleus or ion to refer to the nucleus plus inner-shell electrons.

[6]For an interesting and detailed study on the possibility of detecting vibrational and rotational modes in molecules induced by DM scattering in a gaseous target, see Ref. [58].

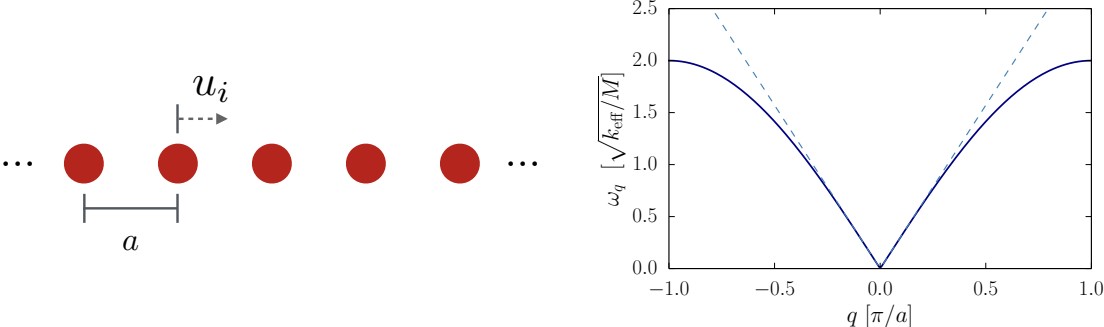

Figure 9: (**left**) A 1D lattice of atoms of mass $M$ has a single longitudinal acoustic phonon branch. (**right**) The dispersion relation of the acoustic phonon is shown over the first Brillouin Zone; near $q = 0$, the phonon has a linear dispersion with slope given by the speed of sound.

where the ... are possible higher order terms in the displacements. Those terms could lead to three-phonon couplings, for instance.

We will take the continuum limit for this system, or equivalently consider long-wavelength excitations, so that the displacement field $u(x, t)$ is a function of position and time. Writing $\Delta x = a$, the sum over positions $i$ can be replaced by an integral over $x$:

$$H = \int dx \left( \frac{1}{2} \rho \dot{u}^2 + \frac{1}{2} \tilde{k}_{\text{eff}} (\nabla u)^2 \right), \tag{67}$$

where $\rho = M/a$ is the mass per unit length, $\tilde{k}_{\text{eff}} = a k_{\text{eff}}$, and we have replaced the finite difference with a gradient. Equivalently, introducing a minus sign for the potential, the Lagrangian for the system is:

$$\mathcal{L} = \int dx \left( \frac{1}{2} \rho \dot{u}^2 - \frac{1}{2} \tilde{k}_{\text{eff}} (\nabla u)^2 \right). \tag{68}$$

This describes a free, massless particle with linear dispersion, $\omega_q = c_s |\mathbf{q}| = c_s q$, and speed of sound $c_s = \sqrt{\tilde{k}_{\text{eff}}/\rho} = \sqrt{a^2 k_{\text{eff}}/M}$. The mode is otherwise known as an acoustic phonon, with an energy that goes to zero in the $q \to 0$ limit. This reflects the fact that the acoustic phonon is a Goldstone boson associated with spontaneous breaking of translational symmetry. As $q \to 0$, all atoms are displaced by the same amount and the arrangement is physically equivalent to the original ground state.

The quantized displacement field (phonon) is written in a standard way, in terms of a mode expansion in the interaction picture:

$$u(x, t) = \sum_j \frac{1}{\sqrt{2 N a \rho \omega_{q_j}}} \left( \hat{a}_{q_j} e^{i q_j x - i \omega_{q_j} t} + \text{h.c.} \right) \tag{69}$$

$$= \frac{\sqrt{aN}}{\sqrt{\rho}} \int \frac{dq}{(2\pi)} \frac{1}{\sqrt{2 \omega_q}} \left( \hat{a}_q e^{i q x - i \omega_q t} + \text{h.c.} \right), \tag{70}$$

with creation and annihilation operators $\hat{a}_q, \hat{a}_q^\dagger$ satisfying the commutation relations $[\hat{a}_q, \hat{a}_{q'}^\dagger] = \delta_{q,q'}$. In the first line, we have written the expansion in terms of a discrete sum – this reflects the actual discrete lattice, with $q_j = 2\pi j/(aN)$ for a lattice of length $aN$. In the second line, we have given the continuum limit result. (If the factor of $\sqrt{aN}$ looks funny in the continuum limit, it is because in the QFT convention we typically normalize the creation

and annihilation operators differently, $[\hat{a}_q, \hat{a}^\dagger_{q'}] = 2\pi\delta(q - q') \to V = aN$ when $q = q'$. This would remove the factor.) As an exercise, you can check that plugging the above expansion into the Hamiltonian gives, in the discrete limit:

$$H = \sum_j \omega_{q_j} \left( \hat{a}^\dagger_{q_j} \hat{a}_{q_j} + \frac{1}{2} \right). \tag{71}$$

We have found that the excitations are described by phonons created by the $\hat{a}^\dagger_q$ operator.

The energy eigenvalues $\omega_{q_j}$ can be solved for exactly, see Ref. [59]. The right panel of Fig. 9 shows the exact phonon band structure over the first Brillouin zone (BZ), accounting for the lattice periodicity. In the long-wavelength limit $q \ll \pi/a$, we see the linear dispersion expected for Goldstone modes. The size of the first BZ is set by $\pi/a$; for a typical material $a \sim$ few Å, and so $q \lesssim$ keV in the first BZ. In the $q \to 0$ limit, the propagation speed is the sound speed $c_s$ with typical values are $\sim 3 - 10$ km/s, yielding typical energies

$$\omega_{\text{acoustic}} = c_s q \simeq 8 \text{ meV} \left( \frac{c_s}{5 \text{ km/s}} \right) \left( \frac{q}{500 \text{ eV}} \right) \tag{72}$$

again for small $q$. The physical interpretation is that all $N$ ions in the crystal are oscillating in phase with the same amplitude as $q \to 0$, which must have zero energy. In an anisotropic material $c_s$ may differ along different lattice directions, leading to distinct dispersion relations for the three acoustic modes.

A realistic lattice has more than one atom per unit cell, resulting in additional phonon branches associated with the relative motions of the atoms within the cell. In the continuum limit, we must define fields $u_1(x, t)$ and $u_2(x, t)$ for the displacements and for each momentum $q$ we therefore have two eigenmodes. The exact dispersions for the 1D model are shown in the right panel Fig. 10, where we see now a gapped mode, called the optical phonon. These modes generically correspond to out-of-phase oscillations within a unit cell, with only mild variation in the energy across the BZ. We can understand the energy scale of optical phonons from dimensional analysis: the normal mode frequencies will be proportional to $\sqrt{\kappa/M_I}$ where $\kappa$ is a spring constant and $M_I$ an ion mass. In addition the acoustic branch has a linear dispersion as $q \to 0$, so $\omega_{\text{acoustic}} \sim \sqrt{\kappa/M_I}(qa)$ where $a$ is the lattice spacing. Identifying $a\sqrt{\kappa/M_I}$ with $c_s$, we have

$$\omega_{\text{optical}} \simeq \frac{c_s}{a} \simeq 10 \text{ meV} \left( \frac{c_s}{5 \text{ km/s}} \right) \left( \frac{0.5 \text{ nm}}{a} \right). \tag{73}$$

We can also estimate optical phonon frequencies based on the electrostatic interactions of ions within the unit cell [60], yielding similar values of

$$\omega_{\text{optical}} \simeq \sqrt{\frac{e^2}{M_I a^3}} \simeq 20 \text{ meV} \sqrt{\frac{14 \text{ GeV}}{M_I}} \sqrt{\frac{(0.5 \text{ nm})^3}{a^3}}. \tag{74}$$

The fact that this energy scale corresponds to the kinetic energy of DM with keV-MeV scale masses, and that they can be excited with a wide range of momentum transfers $\lesssim$ keV, makes the optical phonon branch particularly useful for DM detection.

Finally, going to three spatial dimensions, the displacement field becomes a vector field which can be written as

$$\mathbf{u}_{n,j}(t) = \sum_{\mathbf{q},\nu} \frac{1}{\sqrt{2N_{\text{cell}} M_j \omega_{\nu,\mathbf{q}}}} \left[ \mathbf{e}_{\nu,\mathbf{q}j} a_{\nu,\mathbf{q}} e^{i\mathbf{q}\cdot\mathbf{R}^0_{nj}} e^{-i\omega_{\nu,\mathbf{q}}t} + \mathbf{e}^*_{\nu,\mathbf{q}j} a^\dagger_{\nu,\mathbf{q}} e^{-i\mathbf{q}\cdot\mathbf{R}^0_{nj}} e^{i\omega_{\nu,\mathbf{q}}t} \right], \tag{75}$$

where $n$ labels the unit cell and $j$ labels the atom within the unit cell. The operators $a_{\nu,\mathbf{k}}, a^\dagger_{\nu,\mathbf{k}}$ are phonon annihilation and creation operators and the eigenmodes are given by $\mathbf{e}_{\nu,\mathbf{k}j}$ with

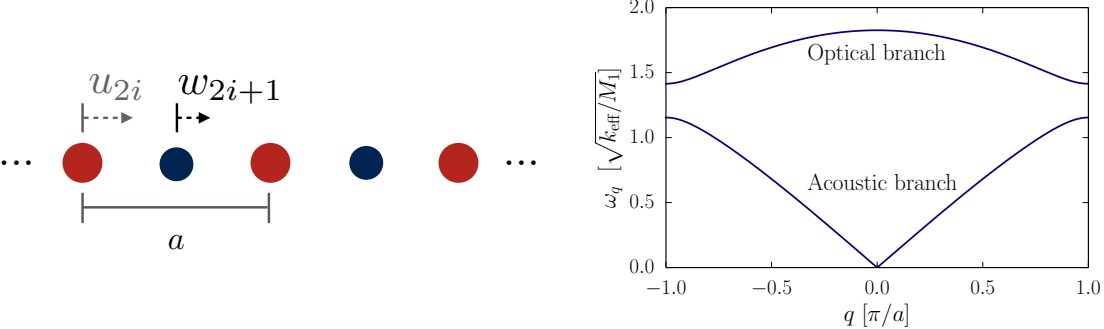

Figure 10: (**left**) A 1D lattice of atoms of mass $M_1$ and $M_2$, with respective displacements from equilibrium $u_i$ and $w_i$. This lattice has both a longitudinal acoustic and longitudinal optical phonon branch. (**right**) The dispersion relations are shown over the first Brillouin Zone for $M_2/M_1 = 1.5$.

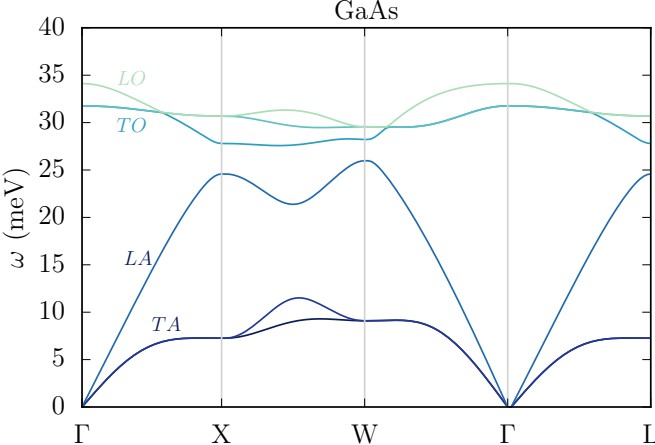

Figure 11: A representative phonon band structure. The $\Gamma$ point is where $\mathbf{q} = 0$, and the acoustic modes are indicated by TA (transverse acoustic) and LA (longitudinal acoustic). The TO and LO branches are the optical phonon modes. Reproduced from Ref. [64].

energy $\omega_{\nu,\mathbf{k}}$. $N_{\text{cell}}$ is the number of unit cells, $\mathbf{R}_{nj}^0$ is the equilibrium position of that atom. Given $n_c$ atoms per unit cell, there are now $3n_c$ phonon branches, labeled by $\nu$. For sufficiently symmetric directions of $\mathbf{q}$, it is also possible to classify the eigenmodes according to transverse modes (displacements perpendicular to $\mathbf{q}$) and longitudinal modes (displacements parallel to $\mathbf{q}$).

With modern density functional theory methods, the eigenmodes and frequencies can all be computed from a first principles approach for a given material, see [61–63] for more details. The band structure for a possible direct detection material (GaAs) is shown in the right panel of Fig. 11, with the phonon energies plotted against a specific path within the first BZ. Here the $q \to 0$ limit is labelled by the $\Gamma$ point, and there are three acoustic modes for a 3-dimensional lattice – two transverse acoustic (TA) branches where the oscillation of the atoms is perpendicular to $\mathbf{q}$ and one longitudinal acoustic (LA) branch. The acoustic modes have linear dispersions for sufficiently small $q$, although the sound speed is different for transverse vs. longitudinal phonons. Because the unit cell for GaAs contains more than one atom, it is seen that there are also optical phonon branches (LO and TO).

## 4.2 Dynamic structure factor

The dynamic structure factor for phonon scattering is given by

$$S(\mathbf{q}, \omega) \equiv \frac{2\pi}{V} \sum_f |\langle f | \sum_I f_I e^{i\mathbf{q} \cdot \mathbf{r}_I} |i\rangle|^2 \delta(E_f - E_i - \omega), \tag{76}$$

where $f_I$ are again the normalized interaction strengths with the ions. The relative scattering strength will depend on the type of ion, and the factors $f_I$ do not factorize out of the structure factor if the system is composed of different types of ions. Thus, in contrast to the electron-excitation dynamic structure factor, for multi-atom target materials there is not a single dynamic structure, but a continuous class of structure factors depending on how the external probe couples to the individual atoms. As will be discussed further later, this allows for additional interesting effects in the material-dependence of DM scattering, as a way to distinguish different DM coupling scenarios. Note that in this work we will restrict to spin-independent DM interaction strengths $f_I$; if there is a spin-dependent interaction potential, then one must also perform an average over all possible spin states of the ions. An extensive review of the dynamic structure factor for phonons, including such spin-dependent interactions, can be found in Ref. [65].

First, it is worth commenting on how the kinematics of single-phonon excitations compares to elastic nuclear recoils. As discussed above, there are two basic branches of phonons we could consider, acoustic and optical phonons. The comparison of DM scattering kinematics with the dispersion relations of these phonons is illustrated schematically in Fig. 8. For the acoustic phonon branch, the energy deposited must be $\omega \sim c_s q$ with $c_s \sim 10^{-5}$. Because the sound speed is so low compared to the DM speed, energy conservation (Eq. 35) leads to the solution $q \sim 2m_\chi v$, which is the same as for sub-GeV elastic nuclear recoils. However, for acoustic phonons the energy deposited will be

$$\omega \sim 2c_s m_\chi v \sim 2\,\text{meV} \times \frac{m_\chi}{100\,\text{keV}}, \tag{77}$$

which is well above the elastic recoil energy for the same DM mass, $E_R \sim 10^{-7} - 10^{-6}$ eV depending on target mass. The energy deposited into a single acoustic phonon could be made larger by using a relatively hard target material, such as SiC [66] or diamond [67] where $c_s \approx 4 - 5 \times 10^{-5}$. However, this energy deposition is still quite small: a DM particle of mass 100 keV has a typical kinetic energy of 100 meV, suggesting that acoustic phonons are not ideal in terms of matching DM kinematics.

The optical phonon branch offers a potential solution to the problem of kinematic matching. Since the dispersions are fairly flat in momentum across the BZ, Eq. 35) gives the approximate solution

$$q \sim m_\chi v \pm \sqrt{(m_\chi v)^2 - 2m_\chi \omega_{\text{optical}}}. \tag{78}$$

The typical energies are around $\omega_{\text{optical}} \sim 30 - 150$ meV, which matches well with the total kinetic energy of DM with mass $\sim$ 10 keV – 1 MeV. The higher energies are also favorable for experimental implementation. From this discussion, we might expect that an ideal target material would likely have a broad spectrum of optical phonon energies in the range of 10 meV up to 150 meV, to allow kinematic matching with a broad range of DM masses. In fact, one can go further and consider systems with some amount of disorder, which further smears out the phonon spectrum and leads to broad spectrum of available modes; this idea was introduced in Ref. [68], which looked at single molecular magnet crystals as a possible direct detection target. Treating the DM coupling to specific modes is more challenging in this type of system, however. In this section, we will study only ordered crystalline lattices, where we can next specify how DM couples to individual phonon branches.

To compute Eq. (76) in terms of phonon excitations, the final states can simply be written by acting with the phonon creation operators introduced in Eq. 75 on the vacuum. For a single phonon being created, $|f\rangle = a^\dagger_{\nu,\mathbf{k}}|0\rangle$, while multiphonon excitations are also possible. In order to detect the phonon excitations being created, the energy deposited is necessarily well above the operating temperature of the experiment, and it is a good approximation to take $T = 0$ and assume $|i\rangle = |0\rangle$ for the initial state. The ion positions are written as $\mathbf{r}_I = \mathbf{R}^0_{nj} + \mathbf{u}_{n,j}$. Substituting this into the exponential appearing in the matrix element gives:

$$f_j e^{i\mathbf{q}\cdot\mathbf{R}^0_{nj}} \exp\left(i\mathbf{q}\cdot\sum_{\nu,\mathbf{k}} \frac{1}{\sqrt{2N_{\text{cell}}M_j\omega_{\nu,\mathbf{k}}}}\left[a^\dagger_{\nu,\mathbf{k}}\mathbf{e}^*_{\nu,\mathbf{k},j}e^{-i\mathbf{k}\cdot\mathbf{R}^0_{nj}} + a_{\nu,\mathbf{k}}\mathbf{e}_{\nu,\mathbf{k},j}e^{i\mathbf{k}\cdot\mathbf{R}^0_{nj}}\right]\right). \tag{79}$$

Expanding this operator will contain a 0-phonon contribution, a 1-phonon creation contribution, and so on. (Note that in Eq. 75 we gave the time-dependent Heisenberg or interaction picture operator $\mathbf{u}_{n,j}(t)$, but the matrix elements given in Eq. 76 are computed with the Schrödinger operators where the time-dependence of the states has already been taken into account in Fermi's Golden rule, leading to the energy-conserving delta function. This is why the time-dependent phase factors have been removed in substituting in Eq. 75.)

To perform the full expansion explicitly in terms of phonon creation and annihilation operators, one can use the Baker-Campbell-Hausdorff formula, and the general result can be found in Refs. [69, 70]. To simplify the discussion, here we just give a qualitative argument for the form of the single-phonon structure factor. First, note that taking only the zeroth-order term in the exponential of phonon creation operators $a^\dagger_{\nu,\mathbf{k}}$, there are no phonon transitions, and this just corresponds to DM elastically recoiling off the lattice as a whole. The leading nontrivial contribution comes from expanding the exponential to linear order, which allows for single-phonon creation. Summing over final states $|f\rangle = a^\dagger_{\nu,\mathbf{k}}|0\rangle$, this leads to the single-phonon structure factor

$$S^{(1-ph)}(\mathbf{q},\omega) = \frac{2\pi}{V}\sum_{\nu,\mathbf{k}}\left|\sum_{n,j} f_j e^{i(\mathbf{q}-\mathbf{k})\cdot\mathbf{R}^0_{nj}}e^{-W_j(\mathbf{q})}\frac{i\mathbf{q}\cdot\mathbf{e}^*_{\nu,\mathbf{k},j}}{\sqrt{2N_{\text{cell}}M_j\omega_{\nu,\mathbf{k}}}}\right|^2 \delta(E_f - E_i - \omega_{\nu,\mathbf{k}}). \tag{80}$$

The *Debye-Waller factor* is $W_j(\mathbf{q}) = \frac{1}{2}\sum_{\nu,\mathbf{k}}\frac{|\mathbf{q}\cdot\mathbf{e}^*_{\nu,\mathbf{k},j}|^2}{2N_{\text{cell}}M_j\omega_{\nu,\mathbf{k}}}$, which roughly speaking accounts for the effect of the zero-point motion of the ions in the lattice.

We next use the fact that for $\mathbf{q}$ smaller than any reciprocal lattice vector $\mathbf{G}$, the sum over lattice sites simply enforces momentum conservation:[7]

$$\sum_n e^{i(\mathbf{q}-\mathbf{k})\cdot\mathbf{R}_n} = N_{\text{cell}}\delta_{\mathbf{q},\mathbf{k}}, \tag{81}$$

since phonon modes are only defined for $\mathbf{k}$ within the first Brillouin Zone. This implies that we will have excitation of any phonon with the same momentum $\mathbf{q}$ and energy $\omega$. The single-phonon structure factor then simplifies to

$$S^{(1-ph)}(\mathbf{q},\omega) = \frac{2\pi N_{\text{cell}}}{V}\sum_\nu\left|\sum_j f_j e^{-W_j(\mathbf{q})}\frac{i\mathbf{q}\cdot\mathbf{e}^*_{\nu,\mathbf{q},j}}{\sqrt{2M_j\omega_{\nu,\mathbf{q}}}}\right|^2 \delta(E_f - E_i - \omega_{\nu,\mathbf{q}})$$

$$\equiv \frac{2\pi}{\Omega}\sum_\nu \frac{|F_\nu(\mathbf{q})|^2}{\omega_{\nu,\mathbf{q}}}\delta(E_f - E_i - \omega_{\nu,\mathbf{q}}), \tag{82}$$

where the second line defines a single-phonon form factor $F_\nu(\mathbf{q})$ and we defined $\Omega = V/N_{\text{cell}}$ as the primitive unit cell volume. This form factor sums over the coupling of the probe with

---

[7]Since reciprocal lattice vectors are defined by the condition $e^{i\mathbf{G}\cdot\mathbf{R}} = 1$, momentum is only conserved up to a reciprocal lattice vector, $\mathbf{k} = \mathbf{q} + \mathbf{G}$. For nonzero $\mathbf{G}$, this is called *Umklapp scattering*.

the ions $f_j$ in the unit cell, multiplied by the normalized motion of that ion $\propto \mathbf{e}^*_{\nu,\mathbf{q},j}/\sqrt{M_j}$ and is therefore describing an effective coupling of the probe with a particular phonon mode, accounting for interference effects. This structure factor therefore describes coherent scattering off the ions in the lattice. However, we explicitly see the 1-phonon form factor is an intrinsic quantity of the material and does not scale with the size of the system.

Assuming the contact interaction limit of Eq. (33) with $M \gg q$ and again applying our main rate formula Eq. 24, the rate is given by

$$R^{(1-ph)}_\chi = \frac{1}{\rho_T} \frac{\rho_\chi}{m_\chi} \int d^3\mathbf{v} f_\chi(\mathbf{v}) \int \frac{d^3\mathbf{q}}{(2\pi)^3} d\omega \frac{\pi \bar{\sigma}_n}{\mu^2_{\chi n}} \delta(\omega + E'_\chi - E_\chi) S^{(1-ph)}(\mathbf{q}, \omega), \qquad (83)$$

where we will take $f_j \to A_j$ for equal couplings to protons and neutrons.

While the eigenmodes and dispersions of the phonon branches must be solved by DFT methods for arbitrary $\mathbf{q}$, it is possible to obtain approximate results in the long-wavelength limit $q \ll \pi/a$ where $\pi/a$ is the typical size of the first BZ. In this limit, we know that the acoustic phonon modes are Goldstone bosons of broken translation invariance, and that as $q \to 0$ all the ions are displaced by the same amount for a zero-energy mode. Comparing with Eq. (75) for the displacement $\mathbf{u}$, we see that in order for the $\sqrt{M_j}$ dependence to drop out, the eigenmodes for the acoustic phonons must be given by $|\mathbf{e}_{\nu,\mathbf{q},j}| = \sqrt{M_j}/\sqrt{\sum_d M_d}$ as $q \to 0$; here the factor of $\sqrt{\sum_d M_d}$ is just to give a normalized eigenvector, where $d$ sums over all ions in the unit cell. In addition, we can restrict only to the longitudinal acoustic (LA) phonon branch where $\mathbf{e}_{LA,\mathbf{q},j} = \hat{\mathbf{q}}\sqrt{M_j}/\sqrt{\sum_d M_d}$ because of the $\mathbf{q} \cdot \mathbf{e}^*_{\nu,\mathbf{q},j}$ dot product. This gives the LA single-phonon form factor in the long-wavelength limit:

$$\lim_{q \to 0} S^{(1-ph,LA)}(\mathbf{q}, \omega) \approx \frac{2\pi}{\Omega} \frac{q^2 |\sum_j A_j|^2}{2(\sum_d M_d) c_s q} \delta(c_s q - \omega), \qquad (84)$$

where we have also taken $e^{-W_j(\mathbf{q})} \approx 1$ and an isotropic speed of sound. The form of this structure factor is similar to that computed for the harmonic oscillator model, $\propto q^2 A^2/(2m_N \omega_0)\delta(\omega - \omega_0)$ for single-phonon excitations. The difference here is that we are taking coherent sum of the couplings over the ions in the unit cell, $|\sum_j A_j|^2$, as well as dividing by unit cell mass. In addition, replacing $\omega_0$ with the linear dispersion of the acoustic phonons leads to a $\sim q$ scaling for the single phonon structure factor, rather than the $\sim q^2$ scaling that was found in the toy harmonic oscillator model. Eq. 84 shows a coherent coupling enhancement over ions in a unit cell for acoustic phonons, and in total scales as the number of ions in the unit cell. As noted above, however, the kinematic matching is not ideal.

We next turn to the optical phonon branch. For DM models where $f_j = A_j$, there is instead a destructive interference for the optical phonon coupling. To see why, let us assume $M_j = A_j m_n$. Then we can rewrite the structure factor as

$$S^{(1-ph)}(\mathbf{q}, \omega) = \frac{2\pi}{\Omega} \sum_\nu \left| \sum_j e^{-W_j(\mathbf{q})} \frac{i\sqrt{M_j}\mathbf{q} \cdot \mathbf{e}^*_{\nu,\mathbf{q},j}}{\sqrt{2m_n^2 \omega_{\nu,\mathbf{q}}}} \right|^2 \delta(\omega - \omega_{\nu,\mathbf{q}}). \qquad (85)$$

In the long-wavelength limit, we can exploit the scaling of the LA phonon mode and rewrite the dot product $\sqrt{M_j}\mathbf{q} \cdot \mathbf{e}^*_{\nu,\mathbf{q},j}$ as a dot product with the LA mode, $\mathbf{e}_{LA,\mathbf{q},j} \cdot \mathbf{e}^*_{\nu,\mathbf{q},j}$. However, by definition for normal modes, the optical phonon eigenmodes are orthogonal to the acoustic phonon eigenmodes, so this dot product vanishes as $q \to 0$. For example, for a unit cell with two atoms, in the $q \to 0$ limit, the LO modes are given explicitly by

$$\mathbf{e}_{LO,\mathbf{q}\to 0,1} \approx \hat{\mathbf{q}} \frac{\sqrt{M_2}}{\sqrt{M_1 + M_2}}, \quad \mathbf{e}_{LO,\mathbf{q}\to 0,2} \approx -\hat{\mathbf{q}} \frac{\sqrt{M_1}}{\sqrt{M_1 + M_2}}. \qquad (86)$$

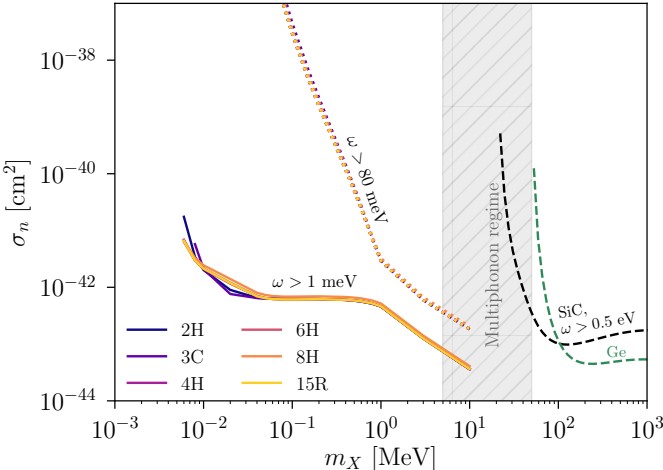

Figure 12: Reproduced from Ref. [66]. Sensitivity of a SiC target to DM-nucleon interactions with a massive mediator. Results are shown assuming kg-year exposure and zero background and for scattering into acoustic phonons ($\omega > $ meV) and for scattering into optical phonons ($\omega = \omega_{LO} \approx 35$ meV). Also shown are example nuclear recoil sensitivities.

Thus, the rate to produce optical phonons in this model (where the DM coupling is proportional to mass) is highly suppressed compared to the acoustic phonon rate, despite the kinematic advantages. This effect was observed in the first calculations of scattering into optical modes for specific materials [64, 71], and shown to be true in general using the orthogonality argument in Ref. [72]. It can be shown instead that the leading behavior of the structure factor scales instead goes as [69, 72]

$$S^{(1-ph,LO)}(\mathbf{q}, \omega) \approx \frac{2\pi}{\Omega} \frac{q^2 A_1 A_2}{2(M_1 + M_2)\omega_{LO}} \frac{q^2 a^2}{16} \delta(\omega - \omega_{LO}). \tag{87}$$

This has a similar form to the previous single-phonon excitation factors derived, but there is an additional $(qa)^2$ suppression when $q \ll \pi/a$ due to the destructive interference in the coupling. The structure factor thus scales as $\sim q^4$ for sub-MeV DM scattering. This behavior has been confirmed in numerical calculations [71, 72], and leads to the reduced cross-section sensitivity for producing a single optical phonon in Fig. 12.

Given that the single-optical-phonon rate scales as $q^4$, it is worth considering whether the 2-phonon contribution to the rate is comparable, since it is expected to have the same scaling. This question was studied in Ref. [69], where it was found that the 2-phonon contribution does indeed scale as $q^4$, but is still smaller than the single-phonon rate, at least for sub-MeV DM.

## 4.3 Dark photon couplings

Up to this point, we have dealt with equal proton and neutron couplings (and zero electron coupling), but it is instructive to also consider a dark photon mediator for single phonon excitations. The scaling for the acoustic and optical structure factors above is not universal and depends on the DM model couplings. The situation is quite different for dark photons, with enhanced couplings to optical phonons and a destructive interference with acoustic phonons. With this example, we will see the possibility of selecting target materials to optimize for a certain DM model.

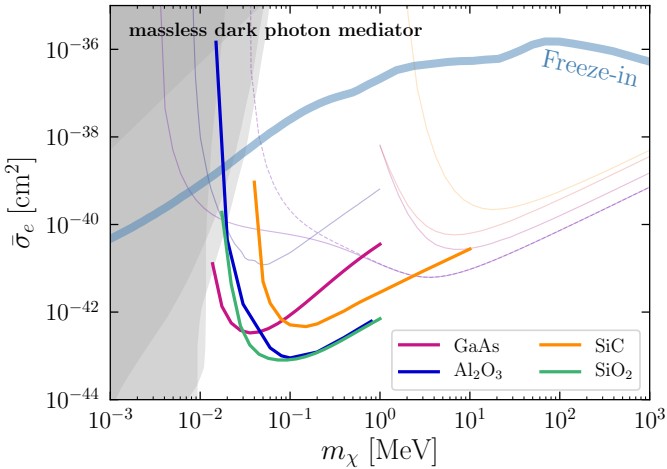

Figure 13: Cross section for 3 events/kg-year for single optical phonon excitations in various polar materials, and assuming a massless dark photon mediator. For this mediator, the convention in the literature is to show projections in terms of the DM-electron cross section $\bar{\sigma}_e$ even when the scattering is into phonons; the mapping is performed by translating the phonon reach in terms of the coupling parameters $\epsilon g_\chi$ and translating into $\bar{\sigma}_e$ for a massless dark photon mediator. This allows comparison with experiments searching for DM-electron scattering, which can probe the same model; the different faint lines in this plot are the various projections for DM-electron scattering in Fig. 7. The thick blue line is the predicted cross section if all of the relic DM is produced by freeze-in interactions [9, 33] and the shaded regions are constraints from stellar emission [8, 50].

For massless dark photon mediators, the DM will couple equally and oppositely to electrons and protons, similar to the ordinary photon, but with an additional overall factor of $\epsilon g_\chi$. The electron coupling introduces some additional complication, since as the ions undergo displacements, the electrons will respond on a rapid time scale. With the same Born-Oppenheimer approximation allowing decoupling of ion and electron motion, the electron response to ion motion can be calculated with first-principles approaches, such that one can determine an effective *dynamical* ion charge. This leads to the definition of the *Born effective charge*, which is the dynamical ion charge in the long-wavelength limit. Formally, it is a charge tensor for each ion $j$ in the unit cell, defined as the change in polarization $\mathbf{P}$ resulting from a displacement to ion $j$:

$$\mathbf{Z}_j^* \equiv \frac{\Omega}{e} \frac{\partial \mathbf{P}}{\partial \mathbf{u}_j}. \tag{88}$$

The Born effective charges are nonzero for polar materials, while they vanish for standard non-polar semiconductors such as Si and Ge. Let us take as a simple example of a polar material GaAs, which has a unit cell of just two ions. The Born effective charges can be approximated to be diagonal and isotropic, so that $\mathbf{Z}_{Ga}^* \approx \mathrm{diag}(2.27, 2.27, 2.27)$ and $\mathbf{Z}_{As}^* \approx \mathrm{diag}(-2.27, -2.27, -2.27)$ [71], describing an effective charge sharing/splitting between the two ions. If one had modeled the Ga as donating all 3 outer shell electrons to the As, the electric charges would be +3 and −3 of the two ions, but the actual Born effective charges of +2.27 and −2.27 account for the deformation of the electron wavefunctions as the ion is displaced.

Since we are dealing with a net neutral target, the sum of Born effective charge tensors for the ions must also be equal to zero. Note that aside from determining the structure factor,

the polarization induced **P** implies that the phonon energies must be re-calculated including the electrostatic energy of this polarization. This leads to an additional contribution to the dynamical force matrix, and an increase of the LO phonon energy. For further discussion of the Born effective charges and their effect on the LO energies, see discussion in Refs. [70,71].

Recalling the out of phase oscillations for LO modes, Eq. (86), and the fact that Ga and As have opposite Born effective charges, we see that the LO mode in a polar material can be thought of as a coherently oscillating dipole in the $q \to 0$ limit. Thus, we can expect that the dark photon mediator primarily couples to the LO mode. To relate the dynamical ion charge to the DM couplings $f_j$, we must further use the fact that the ion charge will be screened in a medium, where the relevant screening factor is given by $\epsilon_\infty$. Here $\epsilon_\infty$ is the long-wavelength dielectric screening at frequencies below the electron band gap but well above the optical phonon frequencies, such that $\epsilon_\infty$ only receives contributions from valence electrons. (At frequencies below the optical phonon frequencies, the optical phonons also contribute to dielectric screening, giving rise to a low-frequency dielectric constant $\epsilon_0$, with $\epsilon_0 > \epsilon_\infty$ in a polar material.) Then, taking $f_j \to Z_j^*/\epsilon_\infty$ and using Eq. (86), we accordingly find

$$S^{(1-ph,LO)}(\mathbf{q}, \omega) \approx \frac{2\pi}{\Omega} \frac{(Z^*)^2}{\epsilon_\infty^2} \frac{q^2}{2\mu_{12}\omega_{LO}} \delta(\omega - \omega_{LO}), \tag{89}$$

where $\mu_{12}$ is the reduced mass of $M_1$ and $M_2$. Because of the opposite signs of both the phonon eigenmodes and the Born effective charges, there is a coherent sum over the ions in the unit cell, in contrast to the case in Eq. (87). We find the same form and $q^2$ scaling as the harmonic oscillator toy model, if we make the identification of the nucleus mass with $\mu_{12}$ and the replacement of the nucleus coupling with $Z^*/\epsilon_\infty$. The structure factor in the case is also sometimes written in terms of a Fröhlich interaction which characterizes electron-phonon interactions, since the interaction of the electron is very similar to that of DM through a dark photon mediator [64,71]. Finally, for the acoustic phonons, the opposite Born effective charges implies a destructive interference when we sum coherently over ions in the unit cell, with the structure factor going to 0 in the limit $q \to 0$.

From these examples, we see that polar materials with large effective charges and a range of optical phonon energies are nearly-ideal target systems for DM interacting through a dark photon, since they enjoy both the kinematic matching and a coherent sum over the ions in the unit cell. For crystals with multiple optical phonon energies, it is also often the case that the highest-energy mode gives the strongest coupling [71]. This is because large effective charges also implies larger electrostatic energies associated with the phonon. For DM which couples equally to protons and neutrons, instead the rate is determined primarily through a combination of the sound speed $c_s$, target nucleus masses, and optical phonon energies of the target system, depending on the energy threshold. Fig. 13 shows cross section sensitivities for example polar materials experiments, including GaAs and $Al_2O_3$ which are planned to be used in experimental collaborations. Studies of additional target materials can be found in Refs. [42,66,71,73,74]. Due to the resonant response, it can also be seen that single phonon excitations can give a much larger rate than DM-electron scattering (faint lines) for sub-MeV dark matter, at least in the materials studied so far; we will explore the DM-electron response more in the following section. Note that in Fig. 13, a massless dark photon mediator has been assumed where $F_{DM}(q) = (\alpha m_e/q)^2$. For sub-MeV DM with $q <$ keV, this form factor can be quite large. For massive dark photon mediators with $F_{DM}(q) = 1$, the rate is much smaller and there is very limited sensitivity to cosmologically interesting parameter space from optical phonon excitations.

While we have mainly taken an isotropic approximation for the dynamic structure factors, another advantage of condensed matter systems is the potential directional dependence in $S(\mathbf{q}, \omega)$. If the DM-phonon couplings or the phonon dispersions are highly anisotropic, this

will lead to a modulation of the DM scattering rate as the Earth (and thus crystal) rotates relative to the typical direction of the incoming DM. The modulation is also sensitive to the DM model details. Combined with the fact that the scattering form factor depends on the DM model, it might be possible to obtain some signal-to-background discrimination in phonon-based detection schemes, or in the case of multiple targets and a positive signal, to deduce information about the DM candidate. The directionality of single-phonon excitation rates is explored further in Refs. [66, 71, 73].

## Acknowledgements

Many thanks to my collaborators and colleagues on this research topic, as well as Yoni Kahn for collaborating on the review article [2], from which these lecture notes borrow extensively. Thanks also to the students at Les Houches for their excellent questions and in particular to Gonzalo Herrera Moreno for providing detailed feedback on the notes. I am supported in part by DOE grant DE-SC0019195 and a UC Hellman fellowship.

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
