# Peer review of "Sub-GeV dark matter models and direct detection"

_SciPost Physics Lecture Notes, doi:SciPost Phys. Lect. Notes 43 (2022)_

## Round 1 · Referee Report · Anonymous (Referee 1) · 2021-10-11

Report

The Les Houches lecture notes by Tongyan Lin focus on the direct
detection of sub-GeV Dark Matter. The latter represents a very active
field of research where understanding the advancements requires
theoretical tools borrowed from solid-state physics. The lecture notes
introduce these concepts in a very pedagogical fashion. Hence, it is
a very welcome document for the Dark Matter community, who are, by and
large, particle physicists and it will be useful to students and
experienced researchers alike. Although sometimes the reference list
is a bit thin, it would also be a considerable challenge to make it
more comprehensive. Since these are lecture notes and not a review, I
feel this is entirely acceptable. I recommend the publication of these
notes in SciPost.

---

## Round 1 · Referee Report · Anonymous (Referee 2) · 2021-11-16

Report

These lecture notes cover a topic that is of high interest in the community. The author is an expert in this subject and provides a concise summary of various models for low-mass dark matter and how to detect them, focusing on direct-detection experiments. The author summarized many of the required calculations and provides nice explanations of the important concepts. There is, quite understandably (and as also admitted by the author), some overlap with an excellent review article and with another set of lecture notes that the author has written, but I feel that these notes provide an excellent and concise introduction to the subject. I recommend publication.

---

## Round 1 · Referee Report · Anonymous (Referee 3) · 2021-11-24

Strengths

1- comprehensive set of lecture notes on a fairly young field of research. 2-provides detailed examples which provide good intuition for relevant parameters and their values.

Report

This set of lecture notes is a valuable document to the field. They are suitably different from the author's TASI lecture notes and provide a complementary set of notes for anyone interested in learning more about the subject of sub-GeV dark matter direct detection.

Requested changes

1- after equation 11, please define e and alpha (they are defined later on in the notes, but this is the first appearance) 2- after eq 20, please define v_0 in the expression for R_chi. N_T may be more intuitive if it is written as N_T~6/m_N~6/(131*1.7e-27 kg) 3-after eq 25, make sure to define e, alpha earlier in the text (see comment 1). 4-before eq. 60, is the use of the dielectric function specific to the dark photon? 5- between eq. 85-85, there may be some confusion with the exclamation point and the factorial symbol. 6-after eq 85, missing "to" in sentence "...is highly suppressed compared TO the acoustic phonon rate..."

---

## Round 2 · List of Changes

* Changes requested by report 3
* Added a few additional references and clarifications (thanks to comments from Gonzalo Herrera Moreno)

---

## Editorial Decision

published